



# Baddeleyite microtextures and U-Pb discordance: insights from the Spread Eagle Intrusive Complex and Cape St. Mary's sills, Newfoundland, Canada.

Johannes E. Pohlner[1,2], Axel K. Schmitt[1], Kevin R. Chamberlain[3], Joshua H. F. L. Davies[4,5], Anne Hildenbrand[1], and Gregor Austermann[1]

[1] Institut für Geowissenschaften, Universität Heidelberg, Im Neuenheimer Feld 234-236, 69120 Heidelberg, Germany
[2] Unit of Earth Sciences, Department of Geosciences, University of Fribourg, Chemin du Musée 6, CH-1700 Fribourg, Switzerland
[3] Department of Geology and Geophysics, University of Wyoming, 1000 E. University Ave., Laramie, WY 82071-2000, USA and Faculty of Geology and Geography, Tomsk State University, Tomsk 634050, Russia
[4] Department of Earth and Atmospheric Sciences, University of Québec at Montréal, 201, Avenue du Président Kennedy, H2X 3Y7, Montreal, QC, Canada
[5] Department of Earth Sciences, University of Geneva, Rue des Maraîchers 13, 1205 Geneva, Switzerland

*Correspondence to*: Johannes E. Pohlner (johannes.pohlner@unifr.ch)

**Abstract.** Baddeleyite ($ZrO_2$) is widely used in U-Pb geochronology, but different patterns of discordance often hamper accurate age interpretations. This is also the case for baddeleyite from the Spread Eagle Intrusive Complex (SEIC) and Cape St. Mary's sills (CSMS) from Newfoundland, which we investigated combining high precision and high spatial resolution methods. Literature data and our own observations suggest that at least seven different types of baddeleyite–zircon intergrowths can be distinguished in nature, among which we describe xenocrystic zircon inclusions in baddeleyite for the first time. Baddeleyite $^{207}Pb/^{206}Pb$ dates from secondary ionization mass spectrometry (SIMS) and isotope dilution thermal ionization mass spectrometry (ID-TIMS) are in good agreement with each other and with stratigraphic data, but some SIMS sessions of grain mounts show reverse discordance. This suggests that matrix differences between references and unknowns biased the U-Pb relative sensitivity calibration, possibly due to crystal orientation effects, or due to alteration of the baddeleyite crystals, which is indicated by unusually high common Pb contents. ID-TIMS data for SEIC and CSMS single baddeleyite crystals reveal normal discordance as linear arrays with decreasing $^{206}Pb/^{238}U$ dates, indicating that their discordance is dominated by recent Pb loss due to fast pathway or volume diffusion. Hence, $^{207}Pb/^{206}Pb$ dates are more reliable than $^{206}Pb/^{238}U$ dates even for Phanerozoic baddeleyite. Negative lower intercepts of baddeleyite discordias and direct correlations between ID-TIMS $^{207}Pb/^{206}Pb$ dates and degree of discordance indicate preferential $^{206}Pb$ loss, possibly due to $^{222}Rn$ mobilization. In such cases, the most reliable crystallization ages are concordia upper intercept dates or weighted means of the least discordant $^{207}Pb/^{206}Pb$ dates.

We regard the best estimates of the intrusion ages to be 498.7 ± 4.5 Ma (2σ; ID-TIMS upper intercept date for one SEIC dike) and 439.4 ± 0.8 Ma (ID-TIMS weighted mean $^{207}Pb/^{206}Pb$ date for one sill of CSMS). Sample SL18 of the Freetown Layered



Complex, Sierra Leone (associated with the Central Atlantic Magmatic Province) was investigated as an additional reference.
For SL18, we report a revised $201.07 \pm 0.64$ Ma intrusion age, based on a weighted mean $^{207}Pb/^{206}Pb$ date of previous and new baddeleyite ID-TIMS data, agreeing well with corresponding SIMS data. Increasing discordance with decreasing crystal size in SL18 indicates that Pb loss affected baddeleyite rims more strongly than cores. Employment of SIMS or mechanical abrasion prior to ID-TIMS analysis may therefore produce more concordant baddeleyite data. We emphasize that the combination of high precision and high spatial resolution dating, along with detailed microscale imaging of baddeleyite, is powerful for
extracting reliable age information from baddeleyite from rocks with a complex post-magmatic evolution.

## 1 Introduction

Baddeleyite, a monoclinic $ZrO_2$ polymorph, is currently one of the most commonly used minerals in U-Pb geochronology, especially for mafic rocks, which are traditionally difficult to date (e.g., Schaltegger and Davies, 2017). It grows most readily during the late stage of igneous crystallization from a silica-undersaturated residual melt, and can co-exist with zircon at
conditions near silica saturation (Heaman and LeCheminant, 1993; Schaltegger and Davies, 2017). Where both minerals co-exist, geochronologists often prefer baddeleyite dates because (1) baddeleyite forms almost exclusively during the late stages of igneous crystallization, facilitating age interpretation, (2) it is rarely inherited from country rock, and (3) it is more resistant to Pb loss than zircon (e.g., Heaman and LeCheminant, 1993), as it remains crystalline even at high radiation doses (Lumpkin 1999). However, small degrees of U-Pb discordance are common in baddeleyite and cannot be eliminated by chemical abrasion
techniques (Rioux et al., 2010). Additionally, crystals are often too small for mineral separation (Söderlund and Johansson, 2002), but can be analyzed in situ by secondary ionization mass spectrometry (SIMS; e.g., Schmitt et al., 2010; Chamberlain et al., 2010) or laser ablation ICP-MS (e.g., Renna et al., 2011).

Various mechanisms have been proposed to explain baddeleyite discordance, leading to contradicting approaches for interpreting ages from discordant analyses. During metamorphism or hydrothermal events, fluids with high $SiO_2$ activity often
cause partial reaction of baddeleyite to polycrystalline zircon. Contributions of such zircon rims can produce discordant baddeleyite analyses due to isotopic mixing (e.g., Heaman and LeCheminant, 1993; Söderlund et al., 2013; Rioux et al., 2010). Strategies for mitigating this problem are selective dissolution of baddeleyite and not zircon prior to ID-TIMS (isotope dilution-thermal ionization mass spectrometry) analysis (Rioux et al., 2010), or employing high spatial resolution methods (SIMS or LA-ICP-MS) to analyze baddeleyite domains free of metamorphic zircon. Other mechanisms for baddeleyite discordance have
been proposed, including alpha recoil (Davis and Sutcliffe, 1985; Davis and Davis, 2018), Pb loss due to fast pathway diffusion (Rioux et al., 2010; Söderlund et al., 2013; Schaltegger and Davies, 2017), and isotopic disequilibrium due to $^{231}Pa$ excess (Amelin and Zaitsev, 2002; Crowley and Schmitz, 2009) or $^{222}Rn$ loss (Heaman and LeCheminant, 2000). For accurate age interpretation of a given sample, the relevant discordance mechanism(s) must be identified, and therefore a more complete understanding of discordance in baddeleyite is needed.





In this study, we present geochronological and microtextural data on early Paleozoic dikes and sills of the Spread Eagle
      Intrusive Complex (SEIC) and Cape St. Mary's sills (CSMS) of the Avalon Zone of Newfoundland. These rocks are variably
      affected by low-grade metamorphism, allowing us to study the effect of alteration on baddeleyite, which is important to
      consider for most rocks that are typically selected for baddeleyite dating. To identify baddeleyite discordance mechanisms, we
      applied U-Pb geochronology by SIMS and ID-TIMS on the same baddeleyite crystals, combined with detailed micro-
petrographic characterization of baddeleyite by scanning electron microscopy (SEM) and in situ SIMS dating. We extend the
      comparison of SIMS and ID-TIMS dating to the essentially unaltered baddeleyite from the Duluth gabbro (sample FC-4b,
      Schmitt et al., 2010) and Freetown Layered Complex (sample SL18, Callegaro et al., 2017). Besides deciphering baddeleyite
      discordance, we present previously undocumented types of baddeleyite–zircon intergrowths and document potential pathways
      for common Pb incorporation in baddeleyite.

**2 Regional geology**

      The Avalon Peninsula consists of rocks that were formed as part of the microcontinent Avalonia during the Neoproterozoic
      and early Paleozoic (e.g., Williams, 1979; Murphy et al., 1999; Figure 1). The Cambrian Adeyton and Harcourt groups
      (Hutchinson, 1962; King, 1988; Figure 2), which unconformably overlie Precambrian rocks, consist of well-preserved marine
      sediments with intercalated pillow basalts and mafic tuffs. The feeder pipes or conduits of these volcanic rocks make up a
mafic intrusive complex, called the "Spread Eagle Gabbro" (McCartney, 1967) or "Spread Eagle Gabbro and equivalents"
      (King, 1988). To avoid confusion, we define this unit, consisting of at least eleven dikes, as the Spread Eagle Intrusive Complex
      (SEIC). Lower Ordovician sedimentary rocks are exposed only on Conception Bay islands northeast of the study area (King,
      1988). They are roughly coeval with Avalonia's presumably early Ordovician separation from Gondwana (e.g., Cocks et al.,
      1997; Murphy et al., 2006; Linnemann et al., 2008, 2012; Pollock et al., 2009, 2012). The Cambrian pillow basalts and their
feeder pipes are confined to the western Avalon Peninsula, which experienced deformation and pervasive low-grade
      metamorphism during the Acadian Orogeny (McCartney, 1967), lasting from ca. 420–360 Ma (van Staal and Barr, 2012;
      Willner et al., 2014).

      **2.1 Spread Eagle Intrusive Complex (SEIC) and Cambrian volcanic rocks**

      Cambrian shales of the Chamberlain's Brook Formation and Manuels River Formation contain pillow basalt flows and/or
basaltic metatuffs in five different localities (McCartney, 1967; Greenough, 1984; Greenough and Papezik, 1985b; Fletcher,
      2006). Additionally, our field observations suggest their presence also within the overlying Elliot Cove formation (sensu stricto
      King, 1988). Several lines of indirect field evidence substantiate that these volcanic rocks were fed by the SEIC dikes or pipes
      which crop out in their vicinity (Greenough, 1984; Greenough and Papezik, 1985b). The SEIC intrusions are arranged in a N-
      S trending array (Figure 1), and are usually subcircular pipes with diameters of several hundreds of meters (McCartney, 1967).
Petrologically, they are internally differentiated, ranging from melanogabbros to leucogabbros (Greenough, 1984; Greenough
      and Papezik, 1985b) and monzonites (this study). These subvolcanic rocks are considerably altered to a greenschist facies



assemblage, but better preserved than the volcanic rocks, which almost completely lack primary minerals (Greenough and Papezik, 1985a,b). Geochemical features of the SEIC rocks are similar to rift basalts, suggesting that Avalonia experienced extensional tectonics in the Cambrian (Greenough and Papezik, 1985b; Greenough and Papezik, 1986a; new whole-rock major

and trace element data are given in the supplements). Early attempts at radiometric dating of the SEIC failed, as alteration hampered Rb-Sr whole-rock methods and zircons were not found (Greenough, 1984). Samples of the current study were collected from several SEIC feeder pipes (Table 1).

## 2.2 Cape St. Mary's sills (CSMS)

In addition to the occurrence of the Cambrian igneous rocks described above, the Cambrian succession is intruded by the

CSMS in the southwestern Avalon Peninsula, especially in the upper Cambrian Gull Cove Formation (Fletcher, 2006). The sills are up to 60 m thick and consist mostly of gabbro, but the thickest sills also include up to 3 m thick granophyric dikes and pockets (Greenough and Papezik, 1986b; Fletcher, 2006). The gabbros and granophyres are both unusually rich in hydrous minerals, notably amphibole and biotite, and volatile complexing was probably an important differentiation process of the subalkaline parental magma (Greenough and Papezik, 1986b). Trace element patterns indicate that the magma was generated

in a mantle source that had been metasomatically enriched by subduction zone-derived fluids (Greenough et al., 1993; supplementary data of this study). A CSMS granophyre was the first terrestrial rock for which baddeleyite dating was performed (Greenough, 1984 and pers. comm.). These ID-TIMS analyses of multiple-crystal aliquots yielded an early Silurian U-Pb discordia upper intercept date of $441 \pm 2$ Ma, but all analyses are discordant to 2.0–3.5% (Greenough et al., 1993). Combining this date with a paleolatitude of $32°$ S $\pm 8°$, the CSMS were interpreted as the result of an igneous event after

Avalonia's separation from Gondwana, but before complete closure of the Iapetus (Hodych and Buchan, 1998). Samples of the current study were collected from a 60 m thick sill at the southwestern coast of Lance Cove (Table 1).

## 3 U-Pb geochronology methods

The occurrence and intergrowth of baddeleyite with other phases in SEIC and CSMS was investigated by SEM on polished thin sections, relying on backscattered electron (BSE) imaging and energy dispersive X-ray spectrometry (EDS) with

additional information from cathodoluminescence (CL) imaging. When baddeleyite and/or zircon crystals were too small for mineral separation (<50 µm), they were analyzed in situ in thin section by SIMS for U-Pb dates (Table 1). For samples with larger crystals, polished epoxy grain mounts were prepared from handpicked separates, followed by SEM imaging and SIMS analysis. Every SIMS session was followed by re-imaging of the analysis craters by SEM to identify analyses with contributions from adjacent phases. Selected crystals were removed from grain mounts with a fine needle for single crystal

ID-TIMS analyses.

SIMS analyses were performed using a CAMECA ims1280-HR ion probe at Heidelberg University. Oxygen flooding of the sample chamber was employed to mitigate crystal orientation effects and improve secondary ion yields (Schmitt et al., 2010; Chamberlain et al., 2010; Li et al., 2010), using an oxygen pressure of $2.0 \times 10^{-5}$ to $3.0 \times 10^{-5}$ mbar. The primary ion beam was



focused to a diameter of about 10–15 μm. In cases where an even higher spatial resolution was needed, the field aperture of
the secondary beam was closed to a square of 5–8 μm. For baddeleyite and zircon analyses, analytical procedures were adapted
from Schmitt et al. (2010). The U/Pb relative sensitivity calibration (RSC) against the $UO_2/U$ ratio accounts for differences in
Pb ionization caused by spot-to-spot differences in sputtering behavior. For this, Phalaborwa baddeleyite (Heaman, 2009) was
always used as primary reference material. For grain mount sessions, FC-4b baddeleyite (Schmitt et al., 2010) was included
as secondary reference material. Zircon analyses were calibrated using the reference materials AS3 (Schmitz et al., 2003) for
U-Pb ages and 91500 (Wiedenbeck et al., 2004) for U concentrations. Zirconolite from sample FP7G was analyzed to
investigate the influence of its matrix on U-Pb baddeleyite dates when the primary ion beam overlaps onto both minerals.

For ID-TIMS analyses of samples FP6D and S2E, baddeleyite dissolution and chemistry were adapted from Rioux et al. (2010).
Baddeleyite crystals were plucked from the SIMS grain mount, spiked with a mixed $^{205}Pb/^{233}U/^{235}U$ tracer (ET535) and
dissolved in HCl acid. Solutions were pipetted into beakers to separate them from undissolved zircon domains. Pb and $UO_2$
from baddeleyite were loaded onto single rhenium filaments with silica gel without ion exchange cleanup. Isotopic
compositions were measured on a Micromass Sector 54 TIMS at the University of Wyoming. Common Pb corrections of
SIMS and ID-TIMS analyses were made using the model of Stacey & Kramers (1975) at 400 Ma. The decay constants and
$^{238}U/^{235}U$ ratio are from Steiger and Jäger (1977). Concordia coordinates and uncertainties were calculated using IsoplotR for
SIMS (Vermeesch, 2018) and PBMacDAT for ID-TIMS (Ludwig, 1988).


## 3.1 Secondary reference baddeleyite from the Duluth gabbro and Freetown Layered Complex (FLC)

Reference baddeleyite FC4b is from the anorthositic series of the Duluth Complex, part of the Middle Proterozoic (ca. 1.1 Ga)
North American Midcontinent Rift system (Paces and Miller, 1993). The sample was collected from the anorthositic series of
the complex, and has been described as an olivine-phyric gabbroic anorthosite (Hoaglund, 2010). FC4-b baddeleyite has
yielded dates of 1096.84 ± 0.45 Ma ($^{206}Pb/^{238}U$; all uncertainties stated are 2σ) and 1099.6 ± 1.5 Ma ($^{207}Pb/^{206}Pb$) by ID-TIMS
analysis (Schmitt et al., 2010). Our new SIMS data for FC-4b are from baddeleyite crystals with petrographic properties
comparable to those of Schmitt et al. (2010).

Sample SL18 is an olivine gabbronorite from the Freetown Layered Complex (FLC) in Sierra Leone, which is part of the
Central Atlantic Magmatic Province (CAMP). SL18 consists of plagioclase and augite with minor olivine and accessory
baddeleyite and apatite. Large baddeleyite crystals (with U contents of 1-4 ng) produced a weighted mean $^{206}Pb$-$^{238}U$ date of
198.777 ± 0.047 Ma by ID-TIMS (Callegaro et al., 2017), but these data show some scatter and the mean date was generated
by 7 analyses out of a total of 11 analyses.

Furthermore, this baddeleyite date is significantly younger than all zircon U-Pb ID-TIMS dates from CAMP samples
(Blackburn et al. 2013; Davies et al. 2017) at ~201.5 Ma. However, the $^{207}Pb/^{206}Pb$ date for the SL18 sample is 201.19 ± 0.69
Ma, overlapping with the Ar-Ar dates from the FLC and U-Pb dates from CAMP samples worldwide. The young and slightly
scattered $^{206}Pb$-$^{238}U$ date with an older "CAMP"-type $^{207}Pb/^{206}Pb$ age suggests that SL18 baddeleyite may have been affected
by Pb loss. Callegaro et al. (2017) discussed different age interpretations of SL18 extensively but were unfortunately unable



to determine a robust U-Pb crystallization age. We present additional single crystal ID-TIMS data of SL18, obtained with exactly the same methodology as for Callegaro et al. (2017) at the University of Geneva, but with baddeleyite crystals from

the same separate that are 10–30 times smaller.

## 4 Petrography

### 4.1 Spread Eagle Intrusive Complex (SEIC)

The SEIC rocks have well-preserved igneous textures, but major and accessory minerals are frequently replaced by parageneses indicative of alteration and low-grade metamorphism. Grain size and colour index can vary considerably, ranging from fine-

grained melanogabbros to coarser-grained monzonites and monzosyenites, which are almost pegmatoidal in rare cases. Plagioclase is always completely altered to albite. Many samples also contain large amounts of K-feldspar. In the less potassic samples, K-feldspar is concentrated in interstitial areas or leucocratic pockets, often together with accessory minerals. Minor quartz is often present in baddeleyite-bearing and baddeleyite-free rocks, but usually in secondary pockets and veins. Clinopyroxene is replaced by chlorite to a variable extent, often forming pseudomorphs. Some samples have essentially

unaltered clinopyroxene, but also contain chlorite pseudomorphs. Ilmenite is largely replaced by titanite ± rutile ± magnetite. Other common secondary minerals are calcite, epidote, prehnite and/or pumpellyite, and accessory sulphides (pyrite, chalcopyrite, galena, sphalerite). Apatite is ubiquitous.

All samples contain Zr-bearing accessory phases. Baddeleyite occurs in many samples, usually as <20 μm long, lath-shaped euhedral crystals. Crystals 20–80 μm in length occasionally occur in some samples, and FP6D is the only sample with large

(50–200 μm) baddeleyite crystals (Figure 3b–d). SEIC baddeleyite is commonly intergrown with zircon, forming a variety of textures. The most common case is that baddeleyite crystals contain zircon domains mostly at their rim, but commonly penetrating into the core. This largely pseudomorphic replacement texture is accompanied by feather-like zircon coronas around the crystal (e.g., Figure 4d). Baddeleyite preservation tends to be better if the rock is less altered, but also if crystals are large, as the presence of rather modest zircon overgrowths in the strongly altered monzonite FP6D indicates. Baddeleyite

inclusions in K-feldspar usually lack zircon intergrowths. In samples FP1F and FP1I, clusters of baddeleyite needles are enclosed by zircon crystals (Figure 3a). The enclosing zircon is sometimes almost euhedral, but often with feather-like zircon overgrowths. A special feature we identified in FP12A is that some baddeleyite crystals have zircon inclusions up to 3 μm wide and at most 12 μm long (Figure 4f–k). Secondary zircon overgrowth on the enclosing baddeleyite was rarely observed. Besides that, FP12A contains prismatic euhedral baddeleyite crystals essentially free of zircon, but with a notch on one prism

plane that penetrates into the crystal's core (Figure 4e, l). Zircon crystals without baddeleyite intergrowth are sometimes present also in baddeleyite-bearing rocks. Some of these crystals have an amoeboid surface and a scarred interior. Baddeleyite-free rocks are often either melanocratic or rich in quartz. They contain zircon as the only Zr-bearing mineral, forming euhedral (≤ 20 μm) or anhedral (≤ 50 μm) crystals. Only one sample (FP7G) contains zirconolite $CaZrTi_2O_7$ (all mineral formulae given as stochiometric end-members from Anthony et al., 2001) and rare pyrochlore $(Ca,Na)_2Nb_2O_6(OH,F)$ in addition to baddeleyite





and zircon. Zirconolite occurs mainly in the vicinity of baddeleyite crystals of similar size and form (Figure 3f). It sometimes

has an altered appearance and/or significant Si contents. In rare cases it is partly replaced by titanite + zircon (Figure 3g).

### 4.2 Cape St. Mary's sills (CSMS)

CSMS gabbros are less altered than those of the SEIC. Sample S2B represents a typical CSMS gabbro regarding major phase

mineralogy (clinopyroxene, albite, titanomagnetite, partly chloritized biotite, chlorite pseudomorphs) and has ilmenite, Cr

spinel, ilvaite $CaFe^{2+}_2Fe^{3+}Si_2O_7O(OH)$ and sulphides as minor to accessory phases. Baddeleyite is the predominant Zr-bearing

mineral, coexisting with zirconolite and minor zircon. Habits of baddeleyite vary from euhedral to anhedral, and needle-shaped

($2 \times 300$ μm) to prismatic ($10 \times 20$ μm). Zircon rims are rare.

In contrast to the gabbros, granophyres are strongly leucocratic (> 70 vol.-% albite). Albite crystals are typically $1 \times 0.5$ cm

large in the interior of granophyre pockets, whereas the outermost ca. 5 cm of the pockets are somewhat less leucocratic with

smaller albite crystals. Nonetheless, a strong contrast of grain size and mineralogy characterizes the sharp contact between

gabbros and granophyre pockets. Granophyre samples S2C (center of a pocket) and S2E (transition of a pocket center to the

gabbro contact) are largely identical in major phases (albite, clinopyroxene, ilmenite, chlorite, biotite ± Ti-rich hornblende).

However, the accessory mineral assemblages are highly different. In S2C, zircon is the only Zr-bearing phase with contact to

groundmass minerals. Crystals are up to 200 μm large, isometrical and growth-zoned. Shapes vary from euhedral sector-zoned

with radial cracks in the outer zones (Figure 5c) over grain clusters (Figure 5f) to fan-shaped forms (Figure 5i). BSE and CL

intensities are usually inversely correlated (Figure 5f, g). Some crystals have partly microporous textures and contain anhedral

baddeleyite inclusions (mostly < 2 μm, max. $4 \times 10$ μm in size), usually in the outer zones. Sometimes the inclusions seem to

retrace cracks or crystallographical orientations (Figure 5e). In S2E, the pocket interior has large (up to 200 μm) zircon crystals,

but without baddeleyite inclusions, whereas baddeleyite sometimes forms isolated crystals in the groundmass. Towards the

gabbro contact, baddeleyite becomes more and more predominant until the presence of zircon is mostly confined to baddeleyite

replacement textures. Near the gabbro contact, baddeleyite occurs either as euhedral, lath-shaped crystals with dimensions up

to $20 \times 300$ μm (Figure 5j), or as short prismatic crystals up to 30 μm in diameter (Figure 5k). Baddeleyite within albite usually

lacks zircon intergrowths. Contrastingly, most baddeleyite within chlorite pseudomorphs is pseudomorphically replaced by

zircon, sometimes containing baddeleyite relicts, and surrounded by a feather-like zircon corona (Figure 5m). Near the gabbro

contact, zircon with amoeboidal grain boundaries occurs, as well as zirconolite and gittinsite $CaZrSi_2O_7$. Gittinsite has not

been reported from similar rock types before. Textures include gittinsite–zirconolite intergrowths (Figure 5n) and gittinsite–

titanite intergrowths branching along the fissure plains of surrounding chlorite (Figure 5o).



## 5 U-Pb results

### 5.1 SIMS data

The SEIC SIMS data are presented in Figure 6 (summary in Table 2; detailed data in supplementary Tables S1–S6). In situ baddeleyite analyses of sample FP7G yielded weighted mean dates of 529.9 ± 21.4 Ma ($^{206}$Pb/$^{238}$U; all uncertainties specified in the text are 2σ) and 497.8 ± 73.2 Ma ($^{207}$Pb/$^{206}$Pb). For FP12A baddeleyite, the weighted mean dates are 508.2 ± 11.2 Ma ($^{206}$Pb/$^{238}$U) and 546.6 ± 83.6 Ma ($^{207}$Pb/$^{206}$Pb). Many baddeleyite analyses show surprisingly high contents of common Pb. As a general practice, analyses with high common Pb (<90% radiogenic $^{206}$Pb) were excluded from weighted mean calculations.

In the grain mount sessions, FC-4b baddeleyite was analyzed as a secondary reference in addition to the Phalaborwa baddeleyite. Weighted mean $^{206}$Pb/$^{238}$U dates of FC-4b calculated with Phalaborwa reference are 1118 ± 39 Ma (MSWD = 0.63; n = 28), 1101 ± 44 Ma (MSWD = 0.41; n = 29), 1124 ± 56 Ma (MSWD = 0.91; n = 9) and 1117 ± 23 Ma (MSWD = 2.42; n = 18; session with sample S2E). Therefore, in all grain mount sessions, the $^{206}$Pb/$^{238}$U ID-TIMS reference age of FC-4b (1096.84 ± 0.45 Ma, Schmitt et al. 2010) was reproduced within error limits. Likewise, the $^{207}$Pb/$^{206}$Pb dates we obtained

for Phalaborwa (2058.8 ± 0.7 Ma, MSWD = 6.3, n = 254) and FC-4b baddeleyite (1096.0 ± 2.9 Ma, MSWD = 1.1, n = 84) are in good agreement with the ID-TIMS $^{207}$Pb/$^{206}$Pb data (2059.6 ± 0.35 Ma, Heaman, 2009, and 1099.6 ± 1.5 Ma, Schmitt et al., 2010). Because of the consistency of Phalaborwa and FC-4b results, analyses from both reference materials were combined for obtaining the Pb/U relative sensitivity factor. Despite the good agreement of $^{206}$Pb/$^{238}$U dates of the reference baddeleyite, $^{206}$Pb/$^{238}$U dates of baddeleyite from sample FP6D obtained during the same sessions were less reproducible (516.2 ± 21.2 Ma,

531.9 ± 14.1 Ma and 563.4 ± 15.2 Ma), with reverse discordance in sessions two and three (Figure 6b, c). However, $^{207}$Pb/$^{206}$Pb dates of these sessions are consistent (500.8 ± 18.0 Ma, 502.5 ± 8.6 Ma and 484.1 ± 13.5 Ma), yielding a total weighted mean date of 497.0 ± 6.8 Ma (MSWD = 0.75; n = 77). Common Pb contents tend to be lower than in FP7G and FP12A, but are often still significant. Zircon rims on baddeleyite and baddeleyite-free zircon from all SEIC samples yielded a wide range of $^{206}$Pb/$^{238}$U dates from 142–517 Ma (Figure 6d, f). At least for sample FP6D, $^{206}$Pb/$^{238}$U dates become younger with increasing

U contents. Zircon analyses from SEIC samples have mostly high common Pb contents (<90% radiogenic $^{206}$Pb). Zircon inclusions in baddeleyite (FP12A) yielded $^{206}$Pb/$^{238}$U date ranges of 470–733 Ma and 297–607 Ma for baddeleyite- and zircon-based RSC, respectively.

       For CSMS, baddeleyite of sample S2E (Figure 7; Table 2; Table S5) yielded weighted mean dates of 446.6 ± 15.4 Ma ($^{206}$Pb/$^{238}$U; MSWD = 1.44; n = 21) and 436.5 ± 21.2 Ma ($^{207}$Pb/$^{206}$Pb; MSWD = 0.82) from the in situ session. In contrast, the

grain mount session of the same sample yielded 491.0 ± 19.8 Ma ($^{206}$Pb/$^{238}$U; MSWD = 0.45; n = 39) and 425.5 ± 8.7 Ma ($^{207}$Pb/$^{206}$Pb; MSWD = 1.00), showing reverse discordance (Figure 7b). $^{206}$Pb/$^{238}$U zircon dates from grain mounts are in the range of 411–443 Ma with moderate or low common Pb contents, but in situ dates of anhedral zircon in chlorite pseudomorphs are much younger, combined with high U and common Pb contents. Zircon dates from S2C (Figure S9; Table S4) are often younger than S2E baddeleyite, but most analyses show high common Pb.





For SL18, weighted mean dates are 202.5 ± 2.2 Ma ($^{206}$Pb/$^{238}$U) and 182.7 ± 12.5 Ma ($^{207}$Pb/$^{206}$Pb) for the grain mount session
and 201.3 ± 7.2 Ma ($^{206}$Pb/$^{238}$U) and 177.4 ± 65.4 Ma ($^{207}$Pb/$^{206}$Pb) for the in situ session (Figure 9; Table 2; Table S6). The
$^{206}$Pb/$^{238}$U dates from these sessions are in good agreement with the CAMP age of ~201.5 Ma based on worldwide samples
using zircon (Blackburn et al., 2013; Davies et al., 2017).

**5.2 ID-TIMS data**

ID-TIMS analyses of baddeleyite from SEIC (sample FP6D) and CSMS (sample S2E) yielded normally discordant data that
form linear arrays (Figure 8; Table 3). The upper intercept dates are 498.7 ± 4.5 Ma (FP6D) and 437.0 ± 7.9 Ma (S2E). The
weighted mean $^{207}$Pb/$^{206}$Pb date of S2E is 444.1 ± 4.4 Ma (95% confidence; MSWD = 0.82). $^{207}$Pb/$^{206}$Pb dates of FP6D show
scatter beyond uncertainty (Figure 8c). For both samples, there is a direct correlation between the $^{207}$Pb/$^{206}$Pb dates and the
percentage of discordance, leading to negative lower intercepts for linear regressions on concordia plots (Figure 8). Like the
corresponding SIMS analyses, baddeleyite analyses from FP6D and S2E contained significant common Pb (up to 6 pg).
Additional ID-TIMS data for very small baddeleyite crystals of sample SL18 (Figure 9c; Table 3) yielded $^{206}$Pb/$^{238}$U dates that
are clearly younger than those of the larger crystals of SL18 published in Callegaro et al. (2017). Common Pb contents of
SL18 baddeleyite are lower than those of SEIC and CSMS baddeleyite.

**6 Discussion**

**6.1 Occurrence, textures and interrelations of accessory minerals**

Zirconium-bearing accessory minerals in mafic magmas form during late stages of crystallization in more differentiated
interstitial melt (Heaman and LeCheminant, 1993; Schaltegger and Davies, 2017). In our study, abundance and crystal size of
accessory minerals lack a strong correlation with whole rock Zr content (see supplements), but the more coarse-grained
samples tend to contain larger baddeleyite and zircon crystals. Regardless of crystal sizes, baddeleyite and zircon in SEIC and
CSMS rocks form different types of intergrowths. Baddeleyite in mafic rocks is typically of igneous origin, but metamorphic
processes can cause it to react to polycrystalline zircon (Heaman and LeCheminant, 1993). Metamorphic zircon is therefore
expected to be the most common type of zircon intergrown with baddeleyite in SEIC and CSMS rocks, which all show
petrological evidence for low-grade metamorphism. Although probably less common, magmatic zircon overgrowths on pre-
existing baddeleyite can also form during late-stage igneous crystallization due to an increased SiO$_2$ activity in the melt (e.g.,
Renna et al., 2011). Such igneous zircon overgrowths on baddeleyite have rather euhedral crystal faces and straight interfaces
with baddeleyite (Renna et al., 2011). By contrast, metamorphic zircon shares a more irregular crystal interface with
baddeleyite and has an anhedral exterior, described as "raspberry texture" (Heaman and LeCheminant, 1993) or "frosty
appearance" (Söderlund et al., 2013). For SEIC and CSMS, the typical textural features of igneous zircon overgrowth on
baddeleyite are rarely displayed (Figure 5l), whereas features of metamorphic replacement zircon are frequently observed.
Zircon seems to pseudomorphically replace baddeleyite, accompanied by feather-like zircon coronas (e.g., Figure 4d, 5l),



which we interpret as a result of volume enlargement by the addition of silica during metamorphism. The presence of such coronas can therefore help to distinguish zircon with a baddeleyite precursor from primary zircon in altered igneous rocks.

The extent of baddeleyite replacement by zircon in this study often depends on the host minerals. Baddeleyite surrounded by chlorite shows replacement by zircon more commonly than baddeleyite in albite or epidote group minerals, and baddeleyite

inclusions in K-feldspar mostly lack zircon. We attribute this to local variations in the $SiO_2$ activity during metamorphism: the chloritization of pyroxenes liberates large amounts of Si, whereas alteration of alkali feldspars has a neutral Si balance. Si release sometimes also causes replacement of zirconolite by titanite + zircon (Figure 3g), or titanite + gittinsite (Figure 5o). Alteration by fluids with high $SiO_2$ activity causes baddeleyite replacement by zircon, but fluids poor in Si and rich in Ca can induce the opposite effect even in siliceous rocks (Lewerentz et al., 2019). In sample S2C, multiple μm-sized baddeleyite

inclusions are hosted in the outer zones of zircon, which shows porous domains (Figure 5a), cracks (Figure 5c), and high contents of common Pb, which are typical alteration features (e.g., Corfu et al., 2003; Rayner et al., 2005; Aranovich et al., 2017). Whereas other fluid-mediated processes may also be capable of forming secondary baddeleyite inclusions in altered zircon (Lewerentz et al., 2019), the former presence of fluids with high Ca/Si ratio in S2C is likely due to widespread albitization of plagioclase. Previously reported occurrences of secondary baddeleyite inclusions in zircon are from rocks that

experienced high temperature (mostly amphibolite facies) alteration (Barth et al., 2002; Aranovich et al., 2013, 2017; Lewerentz et al., 2019), and experiments reproducing this texture were conducted at 600°C and 900°C (Lewerentz et al., 2019). However, Cape St. Mary's sills have experienced only subgreenschist facies (Greenough and Papezik, 1986b), or at most lower greenschist facies conditions. Hence, secondary baddeleyite inclusions in zircon can also form at low temperatures, and low-temperature reactions of zircon to baddeleyite and vice versa can occur within the same dike.

**6.1.1 Zircon inclusions in baddeleyite**

A peculiar texture in sample FP12A is baddeleyite with zircon inclusions (Figure 4e–k). Most of these baddeleyite crystals lack zircon overgrowth, and the baddeleyite mantle is coherent even if it is as thin as 1 μm. Thus the zircon crystals clearly represent inclusions and are not parts of a metamorphic rim locally penetrating into the crystal interior. In SIMS analysis of these zircon inclusions, the primary beam overlapped onto baddeleyite and zircon, but at least one zircon crystal gave a

$^{206}Pb/^{238}U$ date considerably above the Cambrian intrusion age using either a baddeleyite-based or a zircon-based RSC (Figure 10). As baddeleyite is reasonably expected to record the age of dike intrusion, the older age indicates that the zircon inclusions predate this magmatism, and must therefore be of xenocrystic origin. We explain this by assimilation of zircon-bearing country rock by a hot, low $SiO_2$ activity magma, where zircon is undersaturated and dissolves (see e.g., Boehnke et al., 2013). Slow Zr diffusion in the melt limits the zircon dissolution rate, so that the melt adjacent to the zircon will develop an exponentially

decreasing Zr concentration gradient (e.g., Harrison and Watson, 1983). Hence, partially dissolved xenocrystic zircon will be surrounded by a halo of elevated Zr concentration in the zircon-undersaturated magma. Such a halo of elevated Zr concentrations is a preferential location for baddeleyite nucleation, even if the bulk of the magma remains undersaturated with regard to baddeleyite. Once a nucleus is formed, baddeleyite will grow preferentially where Zr concentration is highest, this is



at the dissolution interface. If a coherent baddeleyite mantle is formed, the zircon xenocryst becomes shielded from further dissolution.

In contrast to the apparent rarity of this texture suggested by the lack of previous reports, we document several examples in sample FP12A. Not all analyses of zircon cores and baddeleyite overgrowths show clearly older U-Pb dates than baddeleyite without this texture. This can be explained if the zircon xenocrysts experienced Pb loss, and/or if they are derived from detrital zircon that only slightly predates the age of dike intrusion. Detrital zircon populations of Ediacaran and lower Cambrian sedimentary rocks of the Avalon Zone span the range of 530–760 Ma (Pollock et al., 2009), being a possible source of xenocrysts that are up to ~250 Ma older than baddeleyite, and of others that are only slightly older. Besides zircon cores, FP12A and all other samples of our study lack any other recognizable xenocrysts or xenoliths. It appears that the remaining minerals of the assimilated country rock became readily resorbed.

It is possible that this texture has been overlooked in the past, as detailed micropetrographic investigation is necessary to detect it. This is especially true for large baddeleyite crystals typically targeted for ID-TIMS analysis. Alternatively, this texture may be in fact rare, as its formation depends on numerous factors:

1) A melt with low $SiO_2$ activity is needed, being zircon-undersaturated, but close to baddeleyite saturation.

2) The country rock must have zircon, but should not be too siliceous, because otherwise zircon would be stabilized and baddeleyite destabilized. If the country rock is only weakly consolidated, zircon liberation is facilitated.

3) High temperatures and low crystal fraction of the magma are necessary to assimilate country rock effectively. However, this speeds up Zr diffusion and zircon dissolution as well, compromising baddeleyite nucleation and growth.

4) If the zircon crystal is small or the relative crystal orientations of zircon and baddeleyite are unfavorable, baddeleyite may fail to enclose zircon before the latter dissolves completely. This may leave a notch on the baddeleyite crystal, such as in Figures 4e and 4k.

Because of the drastic consequences of zircon intergrowths for geochronology, it is important to carry out careful micropetrographic investigation of baddeleyite crystals, especially for samples with complex metamorphic histories. At least seven different types of baddeleyite–zircon intergrowths have to be considered (compiled in Table 4), including three types not present in our samples: granular baddeleyite droplets can rim zircon that decomposed to baddeleyite + $SiO_2$ as a result of impact shockwave heating (e.g., El Goresy, 1965; Wittmann et al., 2006). The inversion of this reaction was found in a shergottite sample, where primary baddeleyite is often partially rimmed by polycrystalline zircon in the vicinity of impact melt (Moser et al., 2013; Darling et al., 2016). This is the only baddeleyite–zircon intergrowth type where baddeleyite is affected by shockwave metamorphism (Moser et al., 2013; Darling et al., 2016). Besides that, feather-like polycrystalline baddeleyite reaction rims on mantle-derived zircons in kimberlites were found to be the result of desilicification reactions (Kresten, 1973). These seven intergrowth types can occur in combination, complicating textural interpretation. Dating by SIMS or LA-ICP-MS provides the high spatial resolution that is required to unravel the age relationships of baddeleyite–zircon intergrowths. Alternatively, dissolution in hydrochloric acid alone may avoid including zircon domains in ID-TIMS baddeleyite analyses (e.g., Rioux et al., 2010).



## 6.2 Challenges in baddeleyite geochronology by SIMS

Despite many examples of good agreement between SIMS and ID-TIMS data (e.g., the SIMS $^{207}$Pb/$^{206}$Pb date agrees with the
ID-TIMS $^{207}$Pb/$^{206}$Pb and upper intercept dates of FP6D, the SIMS in situ dates agree with both ID-TIMS dates of S2E, and
all dates of SL18 and FC-4b agree), some SIMS sessions yielded dates that deviate significantly, notably in case of grain mount
sessions and/or $^{206}$Pb/$^{238}$U dates. Although SIMS dates of baddeleyite are often not less accurate than ID-TIMS dates (see Sect.
6.3 and 6.4), baddeleyite poses analytical challenges that are largely specific for in situ methods. These are beam overlap on
adjacent phases, possible bias in the U-Pb relative sensitivity calibration (RSC) and unusually high common Pb contents (the
latter also applies to ID-TIMS analyses).

Small crystal sizes of baddeleyite (< 10 μm) often result in primary beam overlap on adjacent phases during SIMS sessions.
This does not necessarily affect the accuracy of baddeleyite dates if the adjacent minerals are U- and Pb-free (e.g., chlorite),
but otherwise, especially in case of intergrowths with zircon and zirconolite, accuracy of baddeleyite dates can be severely
affected. But even so, estimations of the baddeleyite crystallization ages are possible if the extents of potential resulting
inaccuracies can be assessed. This requires knowledge about (1) the approximate U-Pb crystallization ages of baddeleyite and
the contaminating phase – we suppose that the zircon rims of SEIC samples formed at ca. 400 Ma during the Acadian Orogeny,
being ca. 20% younger than SEIC baddeleyite, and that some of the zircon rims experienced Pb loss; (2) possible differences
in U content of the involved minerals, where baddeleyite tends to have the same (this study) or higher (Heaman and
LeCheminant, 1993) U contents than zircon, and lower U contents than zirconolite (Rasmussen and Fletcher, 2004); and (3)
matrix effects that lead to different U-Pb relative sensitivities. As observed from the RSC, baddeleyite $^{206}$Pb/$^{238}$U dates shift to
younger ages when computed against a zircon RSC (–54 to +3% total, –23% average), and zircon dates become older vice
versa (+1 to +121%, +43% average). Similar bias results from baddeleyite–zirconolite beam overlap, although presently this
bias cannot be quantified due to lack of well-characterized zirconolite reference materials. Notably, when SIMS baddeleyite
analyses contain metamorphic zircon contributions, their younger age is partly compensated by the RSC bias, so their influence
on data quality is less severe than in cases where zircon is coeval or older than baddeleyite.

Precision and accuracy of SIMS $^{206}$Pb/$^{238}$U dates strongly depend on the quality of the RSC. Variable degrees of reverse
discordance in different SIMS grain mount sessions of the same sample most likely reflect difficulties in quantification of
matrix effects. RSC accuracy depends on numerous factors, but crystal orientation effects have been long recognized as
particularly important for SIMS U-Pb dating of baddeleyite (Wingate and Compston, 2000). Although oxygen flooding of the
sample chamber proved to be effective for reducing crystal orientation effects, residual bias remains (Schmitt et al., 2010; Li
et al., 2010). In case of grain mounts, tabular crystals will be preferentially oriented with their c axis parallel to the sample
surface. This may be a cause of matrix mismatch in grain mount analyses, whereas in situ mounts, which lack reverse
discordance in this study, are expected to have more random distributions of crystal orientations. Analysis of a sufficiently
large number of crystals (> ca. 25), and randomizing crystal orientations of grain mounts, are advisable strategies for obtaining





more accurate SIMS $^{206}$Pb/$^{238}$U baddeleyite dates. Alternatively, $^{207}$Pb/$^{206}$Pb dates can be used because they are independent from the RSC.

Ideal U-Pb mineral geochronometers exclude common Pb during crystallization, a behavior that baddeleyite, zircon and zirconolite approximately fulfill (e.g., Heaman and LeCheminant, 1993; Rasmussen and Fletcher, 2004). However, in our samples, all three minerals often contain abundant common Pb. Adjacent phases, surface contamination (SIMS), laboratory

blank (ID-TIMS) and mineral inclusions (SIMS and ID-TIMS) can be external sources of common Pb, but steady $^{204}$Pb counting rates even in some SIMS analyses with the spot entirely on baddeleyite demonstrate that some amount of common Pb is intrinsic to these crystals. In case of zircon, it is known that common Pb can be incorporated during alteration (e.g., Watson et al., 1997; Rayner et al., 2005; Geisler et al., 2007). By analogy, the interplay of radiation damage and interaction with fluids may trigger a similar process in baddeleyite and zirconolite. Alteration could also modify the chemical composition

of baddeleyite and may therefore bias the RSC, but a correlation of $^{206}$Pb/$^{238}$U date and common Pb content is detectable only for sample FP7G, which also shows a correlation of common Pb content and the (RSC-independent) $^{207}$Pb/$^{206}$Pb date. Moreover, common Pb contents of baddeleyite in FP7G and FP12A increase with decreasing U content, making it difficult to explain common Pb incorporation with radiation damage, unless U was mobilized as well. Uranium mobilization can cause normal or reverse discordance and may therefore be detectable by further ID-TIMS analyses. Furthermore, baddeleyite and

zircon can be expected to alter differently, as baddeleyite is more resistant to radiation damage (Schaltegger and Davies, 2017). Oxygen isotope analysis by SIMS has been suggested as a tool to detect baddeleyite alteration (Davies et al., 2018), although its routine use is still in its infancy. Zirconolite, which only allows $^{207}$Pb/$^{206}$Pb dating (Rasmussen and Fletcher, 2004), shows high common Pb abundances in sample FP7G (45–93% radiogenic $^{206}$Pb), therefore being unsuited as a geochronometer for altered samples.

**6.3 Baddeleyite discordance**

Although volume diffusion of Pb in baddeleyite is thought to be extremely slow (Heaman and LeCheminant, 2000), a few percent of discordance are often detected in baddeleyite. Different potential discordance mechanisms influence baddeleyite U-Pb systematics in different ways, thus their correct identification is crucial for age accuracy (Figure 11). Naturally observed discordance patterns range from linear and complex arrays to uniform clusters beneath concordia (e.g., Söderlund et al., 2013;

Schaltegger and Davies, 2017; Heaman & LeCheminant, 2000). This means that discordance is not always created by the same mechanism. The challenge is to correctly identify a discordance pattern as a result of a geological process. If contributions from metamorphic zircon rims have caused discordance, a three-component mixing model can be applied, with end-members defined by (1) the igneous crystallization age of baddeleyite, (2) metamorphic formation of zircon, and (3) recent Pb loss of zircon and/or baddeleyite (Söderlund et al., 2013). The oldest $^{207}$Pb/$^{206}$Pb date would then yield a minimum estimate of (1). In

contrast, $^{231}$Pa excess would influence only the $^{207}$Pb/$^{235}$U decay system, making $^{206}$Pb/$^{238}$U ages most accurate (Crowley and Schmitz, 2009), and vice versa in case of $^{222}$Rn loss (Heaman and LeCheminant, 2000). Pb loss due to alpha recoil or fast



pathway diffusion (e.g., along twin-planes) affects both systems similarly, making $^{207}$Pb/$^{206}$Pb ages most accurate (Davis and Davis, 2018).

Metamorphic zircon overgrowth was absent in baddeleyite used for ID-TIMS analysis of S2E and SL18. Even in FP6D, where

it is petrographically evident, the ID-TIMS data appear to be free of a significant isotopic component of metamorphic zircon. This confirms that baddeleyite and zircon can be separated successfully with the method of Rioux et al. (2010), using only hydrochloric acid for dissolution. Consequently, discordance should be attributed to baddeleyite itself instead of zircon intergrowths. Our ID-TIMS analyses show linear arrays that are typical for varying degrees of recent Pb loss. The portion of radiogenic Pb lost by alpha recoil can be predicted based on crystal shapes (Davis and Davis, 2018) and is generally <0.3%

for the crystals of this study. However, many of the ID-TIMS data here indicate that Pb loss exceeds this extent by more than one order of magnitude (Table 3). Hence alpha recoil plays only a subordinate role, depending on the U zonation of the crystals. Fast pathway and/or volume diffusion is therefore likely to dominate baddeleyite discordance in FP6D, S2E and SL18. Intriguingly, the discordia trends for samples FP6D and S2E have negative lower intercepts and show a positive correlation of the $^{207}$Pb/$^{206}$Pb date with the percentage of discordance (Figure 8). We interpret preferential loss of $^{206}$Pb, possibly due to $^{222}$Rn

mobility, to be responsible for this pattern. Our data suggest that this excess $^{206}$Pb loss increases with overall Pb loss, meaning that the least discordant analyses are least affected by this bias. The mechanisms for Pb loss from baddeleyite remain unclear. The radiation dose does not seem to be crucial: except for the in situ session of S2E, negative correlations between U content and $^{206}$Pb/$^{238}$U dates appear to be absent in SIMS and ID-TIMS data of our samples (cf. Söderlund et al. 2013).

Baddeleyite rims appear to be more strongly affected by Pb loss than cores. This is shown by the ID-TIMS data of SL18

(Figure 9c), which show younger $^{206}$Pb/$^{238}$U dates for the smallest crystals, which have a larger surface to volume ratio than the larger crystals. This effect is not as obvious in the SIMS data, owing to much larger uncertainties. Furthermore, ID-TIMS data tend to be more discordant than SIMS data from the same grains (Table 3 vs. Tables S1, S6). The SIMS spots were typically placed in the centers of the grains, but dissolution of the plucked grains for ID-TIMS analysis included the rims as well. Similarly, SIMS $^{207}$Pb/$^{206}$Pb dates tend to be younger than ID-TIMS $^{207}$Pb/$^{206}$Pb dates, possibly due to increased $^{206}$Pb

loss of the rims. Fast pathway diffusion and/or volume diffusion are both possible explanations of intensified Pb loss from the crystal rims, but to differentiate between these processes, both the U zonation within the crystals and the post emplacement thermal history of the sample are not sufficiently known.

## 6.4 Approaches to obtain the most accurate baddeleyite crystallization ages

With Pb loss as a dominant discordance mechanism, $^{206}$Pb/$^{238}$U dates of baddeleyite often underestimate intrusion ages, and

therefore $^{207}$Pb/$^{206}$Pb dates are more accurate. In case of SIMS, another advantage of $^{207}$Pb/$^{206}$Pb dates is their independence from the RSC. This favors the use of $^{207}$Pb/$^{206}$Pb rather than $^{206}$Pb/$^{238}$U ages even for Phanerozoic baddeleyite, where the $^{207}$Pb/$^{206}$Pb date is typically less precise than the $^{206}$Pb/$^{238}$U date. For Mesozoic samples such as SL18, however, comparatively low precisions of SIMS data make it difficult to meaningfully use $^{207}$Pb/$^{206}$Pb dates, although improved $^{207}$Pb/$^{206}$Pb precisions have been achieved with a multi-collection SIMS method (Li et al., 2009). Another limitation for $^{207}$Pb/$^{206}$Pb dates is possible





bias by preferential $^{206}$Pb loss, as indicated by the correlation of $^{207}$Pb/$^{206}$Pb dates with the percentage of discordance (Figure

8). In this case, only the least discordant $^{207}$Pb/$^{206}$Pb dates can be considered reliable. Alternatively, if the excess $^{206}$Pb loss is

proportional to total Pb loss, upper intercept dates are the most accurate. This can be shown for sample FP6D, where the upper

intercept date from ID-TIMS (499 ± 5 Ma) and the $^{207}$Pb/$^{206}$Pb date from SIMS (497 ± 7 Ma) are undistinguishable, but most

of the ID-TIMS $^{207}$Pb/$^{206}$Pb dates are considerably older (504 to 530 Ma, Table 3, Figure 8). If the baddeleyite cores are less

discordant than the rims, the cores should be preferentially targeted for analysis. In the case of ID-TIMS, mechanical abrasion

is potentially helpful in this respect, and there are documented cases where discordance was reduced by mechanical abrasion

(e.g., Corfu and Lightfoot, 1996; cf. e.g., Greenough et al., 1993). A possible alternative to mechanical abrasion is to cut out

baddeleyite cores with a focused ion beam before ID-TIMS analysis (White et al., in review). With SIMS, it is easier to

preferentially target the cores. Hence, SIMS $^{207}$Pb/$^{206}$Pb dates, which are not precise enough to reveal discordance patterns but

potentially more accurate than ID-TIMS, are important as cross-checks, highlighting the power of the combined usage of high

spatial resolution (SIMS) and high precision (ID-TIMS) dating methods.

### 6.5 Intrusion ages of the SEIC, CSMS and FLC

Statements on the intrusion age of SEIC need to consider stratigraphic constraints in addition to the geochronological data. As

the SEIC consists of the feeder pipes of the Cambrian volcanic rocks on the western Avalon Peninsula (Sect. 2.1), the SEIC is

required to be coeval to this volcano-sedimentary succession. Biostratigraphic constraints suggest that the age span of

deposition of the Manuels River Formation roughly equals that of the Drumian stage (Hildenbrand, 2016; Austermann, 2016).

The Drumian stage is currently thought to span from 504.5–500.5 Ma with age uncertainties from these bounds of ca. 2 Ma

(Peng et al., 2012). The occurrence of pillow lavas in the Chamberlain's Brook Formation and Elliot Cove formation expands

the age range of Cambrian volcanism on the Avalon Peninsula to both pre- and post-Drumian (Figure 2). Our SIMS and ID-

TIMS data are thus in good agreement with these stratigraphic limits. We regard the ID-TIMS concordia upper intercept date

of FP6D (498.7 ± 4.5 Ma) as the best available estimate of the intrusion age of the corresponding feeder pipe. As indicated by

the stratigraphic distribution of the volcanic rocks, the other feeder pipes may significantly pre- and/or postdate this age, but

they lack sufficiently precise geochronologic data to establish a firm age range of magmatism. Nevertheless, the improved

perception of the age of SEIC indicates that SEIC magmatism clearly predates the opening of the Rheic Ocean (see Sect. 2).

For the CSMS, zircon from the granophyre sample S2C is considerably altered and the degree of secondary Pb loss is often

very large, limiting its use for geochronology. Baddeleyite from sample S2E, although potentially also altered, better preserves

the igneous age. Our ID-TIMS $^{207}$Pb/$^{206}$Pb baddeleyite date (444.1 ± 4.4 Ma) agrees within error with that of Greenough et al.

(1993). Their sample was derived from the same granophyric dike as ours (Greenough, pers. comm.). Greenough et al. (1993)

analyzed bulk separates of baddeleyite, thus the large number of crystals per aliquot in their analyses may have obscured the

discordance patterns that we observed in our single crystal analyses. Combining the data of both studies, we regard a weighted

mean ID-TIMS $^{207}$Pb/$^{206}$Pb age of 439.4 ± 0.8 Ma (95% conf.; MSWD = 0.94; Figure 12) as the best available estimate of the

intrusion age of the Lance Cove sill, using only the analyses with <3% discordance to minimize bias due to preferential $^{206}$Pb



loss. If this bias is significant even for the least discordant analyses, this $^{207}Pb/^{206}Pb$ date may be an overestimate, but the ID-TIMS upper intercept date of 437 ± 8 Ma, which would be more accurate in this case, does not lead to a better age estimate

due to inferior precision. The date that we report does not rule out the possibility that other sills of Cape St. Mary's are significantly older or younger.

For the FLC, our new ID-TIMS and SIMS data suggest that the intrusive age based on a weighted mean $^{206}Pb/^{238}U$ date reported in Callegaro et al. (2017) is likely too young and reflects some degree of Pb loss. We showed here that smaller baddeleyite crystals from SL18 yield younger ID-TIMS $^{206}Pb/^{238}U$ dates due to more intense Pb loss likely due to fast pathway or volume

diffusion. Therefore, we regard the $^{207}Pb/^{206}Pb$ ID-TIMS date (201.07 ± 0.64 Ma) as the best estimate for the baddeleyite crystallization age of SL18. This age is in agreement with the SIMS dates and all other age constraints from both the FLC and the CAMP (Davies et al., 2017; Callegaro et al., 2017). This new age does not change the overall interpretations of Callegaro et al. (2017).

**7 Conclusions**

A case study of mafic dikes and sills from western Avalon Peninsula, Newfoundland, Canada shows complex textures and age relations for baddeleyite and zircon in mafic rocks that underwent sub- to lower-greenschist metamorphism. Based on new and published microtextural observations, at least seven different types of baddeleyite–zircon intergrowths have to be considered when using baddeleyite as a geochronometer. A previously undocumented type that we discovered in SEIC dikes is xenocrystic

cores of zircon mantled by igneous baddeleyite overgrowths. This study also shows that a combination of high precision and high spatial resolution methods is required to extract reliable age information from baddeleyite. The accuracy of SIMS in situ analyses is affected by calibration bias, possibly due to crystal orientation effects. Unusually high common Pb contents in SEIC and CSMS baddeleyite are probably a consequence of alteration. Baddeleyite discordance detected in ID-TIMS single-crystal analyses is primarily caused by secondary Pb loss, which comprises a component along a zero age intercept discordia,

but also preferential loss of $^{206}Pb$, possibly due to $^{222}Rn$ mobility. Any kind of Pb loss makes $^{207}Pb/^{206}Pb$ ages more reliable than $^{206}Pb/^{238}U$ ages, even for Phanerozoic baddeleyite. Preferential $^{206}Pb$ loss is detectable by direct correlations between $^{207}Pb/^{206}Pb$ dates and degree of discordance. In this case, the most accurate age is either the concordia upper intercept date or the $^{207}Pb/^{206}Pb$ dates of the least discordant analyses. Potential remedies for Pb-loss are to preferentially target the less discordant cores of baddeleyite, either by SIMS, or possibly by applying mechanical abrasion prior to ID- TIMS analysis.


*Author Contributions.* This study is partly based on the MSc thesis of JEP, supervised by AKS. GA and AH applied for funding and helped with fieldwork. SIMS analyses were performed by JEP (SEIC, CSMS), AKS and KRC (SL18). ID-TIMS analyses were performed by KRC (SEIC, CSMS) and JHFLD (SL18). Geochronological interpretations developed from discussions between JEP, KRC and AKS with additions by JHFLD. JEP wrote the manuscript with support from all co-authors.




*Competing interests.* The authors declare that they have no competing interests.

*Acknowledgements.* This project is funded by the Klaus Tschira Foundation (03.131.2017 Förderung von Abschlussarbeiten (M.SC. und B.SC.) in Verbindung mit Projekt 00.272.2015 „Kambrium von Avalonia mit Schwerpunkt Ostneufundland''). 520 Further support came from the German Research Foundation (DFG Scientific Instrumentation and Information Technology programme). KRC was partially supported from Mega-Grant 14.Y26.31.0012 and RNF grant 18-17-00240 of the government of the Russian Federation. Ilona Fin and Oliver Wienand prepared thin sections and epoxy mounts. Robert B. Trumbull (Helmholtz Centre GFZ-Potsdam) contributed whole-rock geochemical data (see Table S7, Figures S10–S12).

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





**Figure 1: Simplified geological map of the western Avalon Peninsula, compiled after King (1988) and O'Brien et al. (2001).**





| System | Series (2019) | Series (2004) | St. Mary's Peninsula | | | Avalon Peninsula | |
|---|---|---|---|---|---|---|---|
| | | | | | | | 489.4 ± 1.9 Ma |
| Cambrian | Furongian | Upper Cambrian | Harcourt Group | Cape St. Mary's sills | Gull Cove Fm. | Elliot Cove fm. | |
| | | | | | Beckford Head Fm. | | ~ 497 Ma |
| | Miaolingian | Middle Cambrian | | Manuels River Fm. | | pillow basalts and mafic tuffs | |
| | | | Adeyton Group | Chamberlain's Brook Fm. | | | ~ 509 Ma |
| | Series 2 | Lower Cambrian | | Brigus Formation | | | |
| | | | | Smith Point Formation | | | ~ 521 Ma |
| | Terreneuvian | | | Bonavista Formation | | | |
| | | | | Random Formation | | | |
| | | | | | | | 541.0 ± 1.0 Ma |

Figure 2: Cambrian stratigraphy of the western Avalon Peninsula, compiled after King (1988), Fletcher (2006) and Austermann (2016). Subdivision of the Cambrian after Peng et al. (2012).





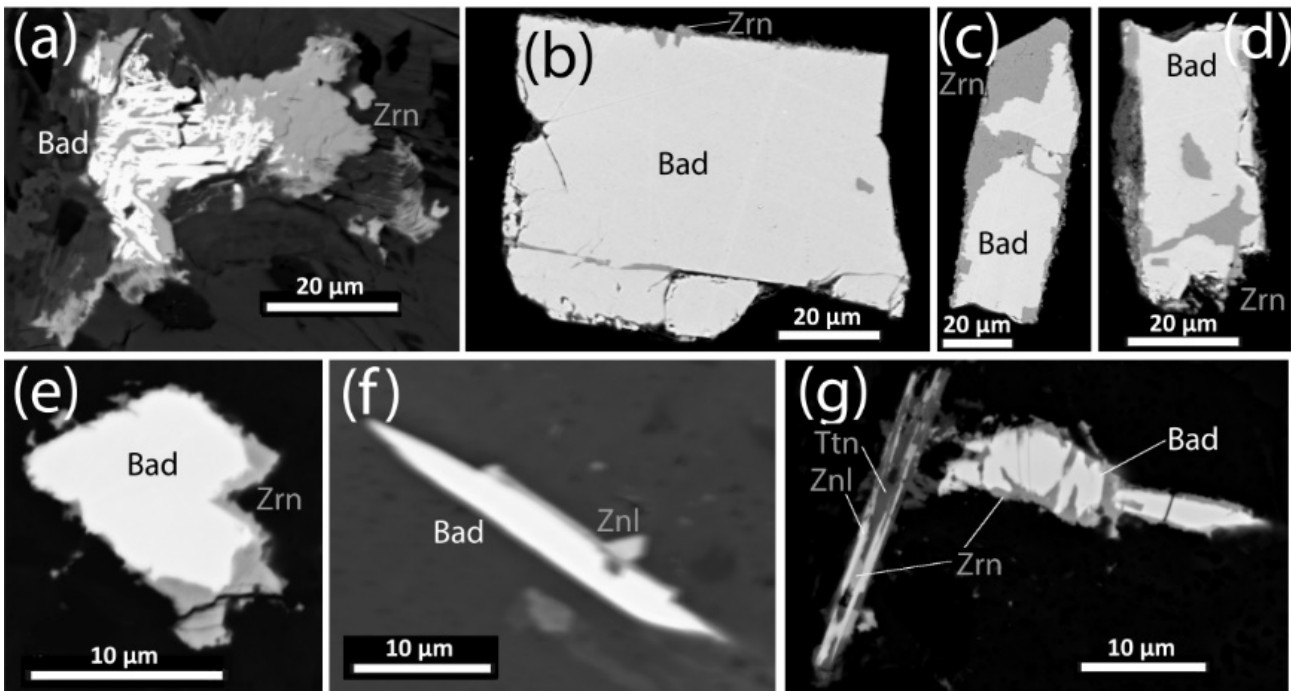

Figure 3: Backscatter electron (BSE) images of accessory minerals in the SEIC. (a) Baddeleyite (Bad) clusters surrounded by zircon (Zrn), sample FP1F. (b-d) Mineral separate of baddeleyite from sample FP6D, showing variable proportions of zircon domains. (e) Baddeleyite with thin zircon rims, sample FP7G. (f) Baddeleyite–zirconolite (Znl) intergrowth, sample FP7G. (g) Baddeleyite and zirconolite, surrounded by zircon and titanite (Ttn).

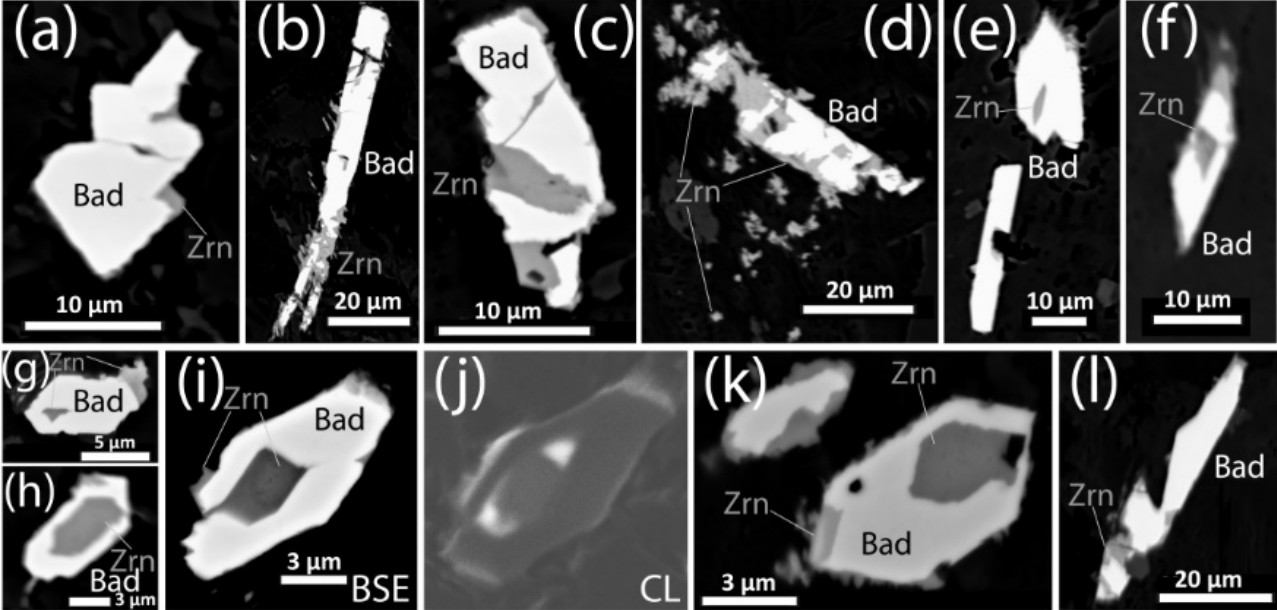

Figure 4: BSE and cathodoluminescent (CL; j) images of baddeleyite-zircon intergrowths in SEIC sample FP12A. (a-d) Baddeleyite of different habits, intergrown with variable proportions of zircon. (e-l) Baddeleyite with zircon inclusions and notches.





**Figure 5: BSE and CL images of accessory minerals in CSMS. (a-i) Different habits of zircon in sample S2C, most of them with baddeleyite inclusions. (j, k, l) Euhedral baddeleyite with and without zircon intergrowth. (m) Relict baddeleyite within zircon pseudomorph, with a feather-like zircon corona. (n) Baddeleyite and zirconolite, intergrown with zircon and gittinsite. (o) Gittinsite-titanite intergrowth within chlorite.**






**Figure 6: SIMS U–Pb baddeleyite and zircon results for SEIC samples. Ellipses in red represent analyses of baddeleyite with zircon inclusions (e) or zircon analyses with <90% radiogenic $^{206}$Pb (d, f). Error ellipses of individual analyses are 1σ whereas the weighted mean ellipse (blue) is enlarged to 2σ.**







Figure 7: SIMS U-Pb baddeleyite (a, b) and zircon (c) results for CSMS samples. Error ellipses of individual analyses (only those used for weighted mean date calculations are shown) are 1σ whereas the weighted mean ellipse (blue) is enlarged to 2σ.





Figure 8: ID-TIMS U-Pb results for samples FP6D and S2E. $^{207}$Pb/$^{206}$Pb dates are less scattered than $^{206}$Pb/$^{238}$U dates, but show a linear correlation with the percentage of discordance. All error bars and ellipses represent 2σ uncertainties.



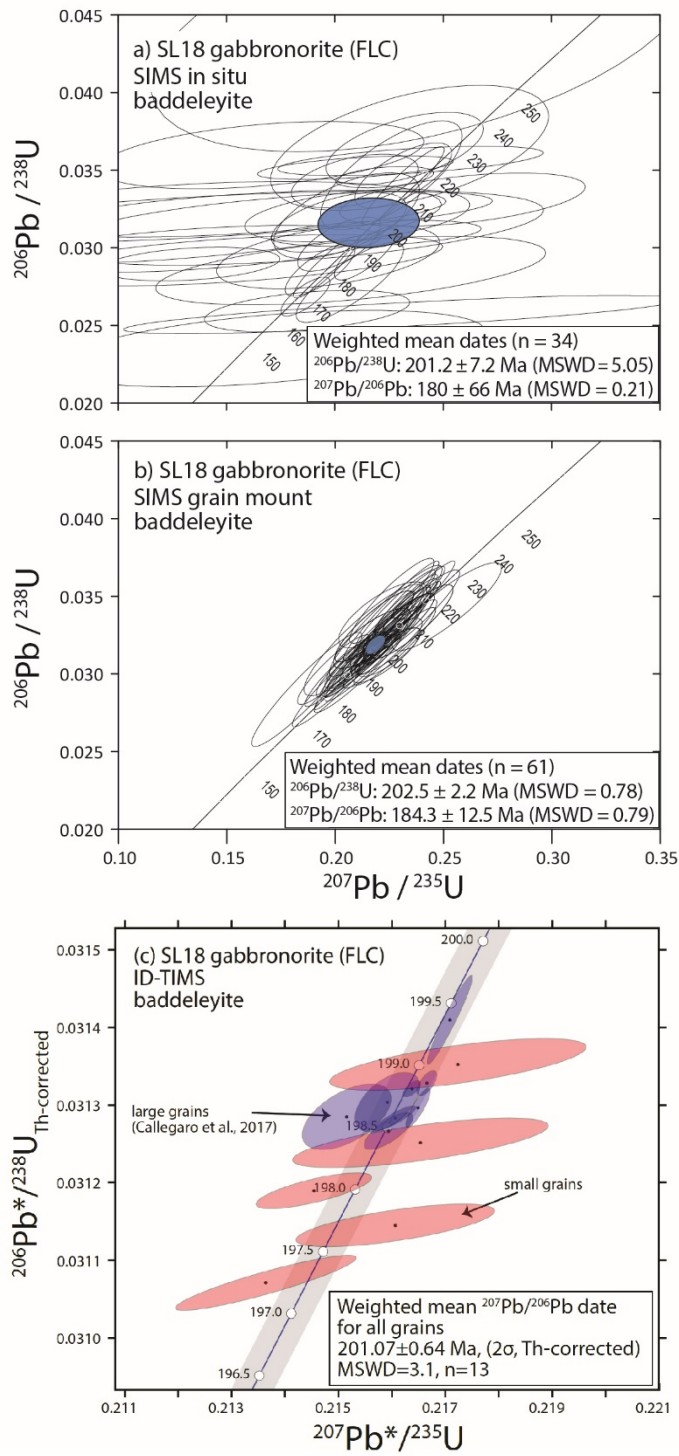

**Figure 9: U-Pb baddeleyite data of sample SL18. For the SIMS analyses (a, b), error ellipses of individual analyses are 1σ whereas the weighted mean ellipse (blue) is enlarged to 2σ. For the ID-TIMS data (c) of large crystals (blue; Data from Callegaro et al., 2017) and small crystals (red; this study), all ellipses represent 2σ uncertainties.**





**Figure 10: Baddeleyite with and without a zircon inclusion, before and after SIMS analysis, with corresponding $^{206}$Pb/$^{238}$U dates (Ma; 2σ uncertainties) calculated using Phalaborwa baddeleyite calibration (black) and AS3 zircon calibration (gray). The fact that the date of the mixed baddeleyite–zircon analysis is older than the other baddeleyite dates for both calibrations indicates a real age difference rather than bias caused by applying a baddeleyite-based calibration for the zircon component of the analysis.**








**Figure 11: Schematic model of the consequences of different discordance mechanism on the isotopic composition of baddeleyite.**








**Figure 12: Re-calculated age for the Cape St. Mary's sills using only the ID-TIMS data of Greemough et al. (1993) and this study which are <3% discordant.**





**Table 1.** Samples used for U-Pb geochronology. *ID-TIMS data for FC-4b are from Schmitt et al. (2010), those for SL18 partly from Callegaro et al. (2017).

| Unit | Sample | Coordinates | Rock type | Typical crystal size (µm) | | SIMS | | ID-TIMS |
|------|--------|-------------|-----------|---------------------------|--|------|--|---------|
| | | | | Baddeleyite | Zircon | In situ | Grain mount | |
| SEIC | FP6D | 47°22.004' N 053°39.802' W | Pegmatoidal monzonite | 50–200 | 50 | | X | X |
| | FP7G | 47°22.098' N 053°39.178' W | Monzogabbro | 5–30 | 5–10 | X | | |
| | FP12A | 47°31,274' N 053°39,259' W | Gabbro | 10–30 | 5–10 | X | | |
| CSMS | S2C | 46°47.756' N 054°05.866' W | Granophyre | Inclusions in zircon | 20–200 | X | | |
| | S2E | 46°47.756' N 054°05.866' W | Granophyre | 5–200 | 50–200 | X | X | X |
| Duluth gabbro | FC-4b | 47°46.118' N 091°21.402' W | Gabbroic anorthosite | 100–200 | 100–200 | X | | X* |
| FLC | SL18 | 08°27' N 13°13' W | Olivine gabbronorite | 10s to 100s | – | X | X | X* |





**Table 2.** Summary of U-Pb SIMS results (see supplementary tables S1-6 for the detailed data).

| Session | n** | $^{206}Pb/^{238}U$ Date (Ma) | ± 2σ | ± 2σ* | MSWD | $^{207}Pb/^{206}Pb$ Date (Ma) | ± 2σ | MSWD |
|---|---|---|---|---|---|---|---|---|
| FP6D gm baddeleyite1 | 24 (20) | 516.2 | 14.9 | 21.2 | 2.03 | 500.8 | 18.0 | 0.54 |
| FP6D gm baddeleyite2 | 30 (30) | 531.9 | 14.1 | | 0.38 | 502.5 | 8.6 | 0.83 |
| FP6D gm baddeleyite3 | 28 (27) | 563.4 | 13.6 | 15.2 | 1.25 | 484.1 | 13.5 | 0.65 |
| sessions 1-3 | | 539.1 | 8.3 | 10.1 | 1.48 | 497.0 | 6.8 | 0.75 |
| FP6D gm zircon | 10 (4) | 142–517 | – | | – | 247–678 | – | – |
| FP7G ins baddeleyite1 | 10 (4) | 544.7 | 46.8 | | 0.39 | 511.3 | 81.8 | 0.43 |
| FP7G ins baddeleyite2 | 16 (7) | 526.0 | 24.2 | | 0.43 | 444.1 | 163.4 | 0.20 |
| sessions 1-2 | | 529.9 | 21.4 | | 0.42 | 497.8 | 73.2 | 0.30 |
| FP7G ins zircon | 1 (1) | 263.8 | 11.5 | | – | 732 | 432 | – |
| FP12A ins baddeleyite1 | 16 (8) | 509.6 | 11.8 | 17.5 | 2.19 | 479 | 111 | 0.39 |
| FP12A ins baddeleyite2 | 6 (4) | 527.5 | 63.0 | | 0.91 | 725 | 160 | 0.54 |
| FP12A ins baddeleyite3 | 6 (4) | 468.2 | 53.6 | | 0.95 | 520 | 264 | 0.05 |
| FP12A ins baddeleyite4 | 5 (2) | 505.7 | 62.6 | 83.0 | 1.76 | 418 | 346 | 1.30 |
| sessions 1-4 | | 508.2 | 11.2 | 13.7 | 1.49 | 546.6 | 83.6 | 0.76 |
| FP12A ins zircon*** | 6 (3) | 311–408 | – | | – | 325–417 | – | – |
| S2C ins zircon | 23 (13) | 426.8 | 26.4 | | 0.46 | 373.6 | 48.4 | 0.88 |
| S2E ins baddeleyite | 24(21) | 446.6 | 12.8 | 15.4 | 1.44 | 436.5 | 21.2 | 0.82 |
| S2E gm baddeleyite | 42 (39) | 491.0 | 19.8 | | 0.45 | 425.5 | 8.7 | 1.00 |
| S2E ins zircon | 5 (1) | 402.0 | 51.2 | | – | 846 | 340 | – |
| S2E gm zircon | 8 (7) | 411–443 | – | | – | 282–443 | – | – |
| SL18 ins baddeleyite | 41 (34) | 201.2 | 3.2 | 7.2 | 5.05 | 180.2 | 66.0 | 0.21 |
| SL18 gm baddeleyite | 61 (61) | 202.5 | 2.2 | | 0.78 | 184.3 | 12.5 | 0.79 |

ins = in situ, gm = grain mount

*2σ uncertainty multiplied with the sqareroot of the MSWD for samples with MSWD > 1

**number in parentheses is without analyses that have high common Pb (<90% radiogenic $^{206}Pb$) or contain contributions from other U-bearing minerals. These analyses are excluded from weighted mean calculations. For zircon, a range of dates is given instead of weighted means, as the crystals may belong to several generations and/or have undergone strongly variable Pb loss

***not including zircon inclusions in baddeleyite. Analyses were acquired during sessions FP12A baddeleyite1–3.





**Table 3.** U-Pb ID-TIMS results of baddeleyite.

| Sample | Weight (µg) | U (ppm) | sample Pb (ppm) | cPb (pg) | Pb* Pbc | Th U | 206Pb 204Pb | 208Pb 206Pb (rad.) | %err | 206Pb/238U (rad.) | %err | 207Pb/235U (rad.) | %err | 207Pb/206Pb (rad.) | %err | 206Pb 238U | ±2σ abs | 207Pb 235U | ±2σ abs | 207Pb 206Pb | ±2σ abs | Rho | discord. (%) |
|---|---|---|---|---|---|---|---|---|---|---|---|---|---|---|---|---|---|---|---|---|---|---|---|
| **FP6D, Spread Eagle Intrusive complex (monzonite)** | | | | | | | | | *Upper intercept date 498.7±4.5 Ma (MSWD 1.7, prob. of fit 0.15)* | | | | *Weighted mean 207Pb/206Pb date = 514 ± 11 Ma (95% conf.; MSWD 5.5, n = 6)* | | | | | | | | | |
| #11 | 1.03 | 1000 | 76.1 | 78.4 | 4.4 | 16.8 | 0.06 | 1155 | 0.07844 | 0.02 | 0.6200 | (0.33) | 0.0573 | (0.27) | 486.8 | 489.8 | 503.9 | 6.0 | 0.57 | 3.52 |
| #15 | 1.13 | 165 | 13.5 | 15.2 | 1.5 | 9.5 | 0.23 | 630 | 0.08005 | 0.08 | 0.6367 | (0.88) | 0.0577 | (0.77) | 496.4 | 500.3 | 518.1 | 17.0 | 0.48 | 4.34 |
| #3 | 1.44 | 414 | 33.9 | 48.9 | 5.7 | 7.8 | 0.15 | 528 | 0.07754 | 0.05 | 0.6156 | (0.47) | 0.0576 | (0.41) | 481.4 | 487.1 | 513.6 | 9.1 | 0.49 | 6.50 |
| #19 | 0.29 | 1456 | 105.5 | 30.4 | 1.0 | 28.8 | 0.13 | 1936 | 0.07585 | 0.04 | 0.6033 | (0.35) | 0.0577 | (0.32) | 471.3 | 479.3 | 517.9 | 6.9 | 0.45 | 9.34 |
| #1 | 0.52 | 662 | 50.2 | 26.3 | 2.3 | 10.5 | 0.16 | 704 | 0.07408 | 0.06 | 0.5932 | (0.56) | 0.0581 | (0.50) | 460.7 | 472.9 | 532.8 | 11.0 | 0.45 | 14.03 |
| #18 | 0.48 | 350 | 26.8 | 12.8 | 2.0 | 5.8 | 0.13 | 400 | 0.07294 | 0.05 | 0.5834 | (0.86) | 0.0580 | (0.81) | 453.8 | 466.6 | 530.0 | 17.7 | 0.45 | 14.88 |
| **S2E, Cape St. Mary's sills (granophyre)** | | | | | | | | | *Upper intercept date 437.0±7.9 Ma (MSWD 0.5, prob. of fit 0.68)* | | | | *Weighted mean 207Pb/206Pb date = 444.1 ± 4.4 Ma (95% conf.; MSWD 0.82, n = 5)* | | | | | | | | | |
| #14 | 0.47 | 370 | 25.2 | 11.8 | 1.6 | 7.0 | 0.03 | 501 | 0.06938 | 0.19 | 0.5299 | (1.01) | 0.0554 | (0.95) | 432.4 | 431.7 | 428.0 | 21.1 | 0.43 | -1.06 |
| #18 | 1.02 | 555 | 37.8 | 38.6 | 3.5 | 10.2 | 0.04 | 713 | 0.06900 | 0.23 | 0.5303 | (0.46) | 0.0557 | (0.38) | 430.1 | 432.0 | 442.0 | 8.4 | 0.57 | 2.78 |
| #33 | 1.31 | 483 | 30.6 | 39.9 | 1.9 | 20.3 | 0.04 | 1409 | 0.06689 | 0.11 | 0.5150 | (0.30) | 0.0558 | (0.27) | 417.4 | 421.8 | 446.0 | 6.0 | 0.46 | 6.62 |
| #28 | 0.64 | 561 | 38.8 | 25.0 | 3.5 | 6.4 | 0.09 | 445 | 0.06603 | 0.33 | 0.5080 | (1.00) | 0.0558 | (0.90) | 412.2 | 417.1 | 444.3 | 19.9 | 0.47 | 7.47 |
| #24 | 0.54 | 209 | 12.9 | 7.0 | 0.9 | 8.1 | 0.05 | 573 | 0.06604 | 0.02 | 0.5097 | (1.27) | 0.0560 | (1.19) | 412.2 | 418.2 | 451.5 | 26.5 | 0.53 | 8.98 |

data produced at the University of Wyoming. Sample Pb: sample Pb (radiogenic + initial) corrected for laboratory blank (0.8 pg). cPb: total common Pb.
Pb*/Pbc: radiogenic Pb to total common Pb (blank + initial). Initial Pb isotopic compositions estimated from Stacey and Kramers (1975) model.

| Fraction | mass U (ng) | sample Pb (pg) | cPb (pg) | Pb* Pbc | Th 206Pb U 204Pb b | 206Pb 204Pb b | 208Pb 238U c | ±2σ % | 207Pb 235U c | ±2σ % | 207Pb 206Pb c | ±2σ % | 206Pb 238U | ±2σ abs | 207Pb 235U | ±2σ abs | 207Pb 206Pb | ±2σ abs | Rho | discord. (%) |
|---|---|---|---|---|---|---|---|---|---|---|---|---|---|---|---|---|---|---|---|---|
| **SL18, Freetown Layered Complex** | | | | | | | | | | | | | *Weighted mean 207Pb/206Pb date = 201.07 ± 0.64 Ma (2σ; MSWD = 2.5, n = 13)* | | | | | | | |
| b1* | 1.40 | 39.6 | 0.39 | 102 | 0.02 | 7073 | 0.031266 | 0.18 | 0.21607 | 0.051 | 0.050142 | 0.13 | 198.574 | 198.63 | 200.6 | 3.1 | 200.6 | 3.1 | 0.88 | 1.04 |
| b2* | 3.64 | 103 | 0.15 | 677 | 0.01 | 46820 | 0.031304 | 0.074 | 0.21637 | 0.033 | 0.050153 | 0.048 | 198.810 | 198.89 | 201.0 | 1.3 | 201.0 | 1.3 | 0.66 | 1.16 |
| b3* | 2.80 | 79.4 | 0.38 | 211 | 0.01 | 14578 | 0.031250 | 0.21 | 0.21594 | 0.075 | 0.050139 | 0.16 | 198.47 | 198.52 | 200.4 | 3.7 | 200.4 | 3.7 | 0.75 | 1.02 |
| b4* | 4.70 | 133 | 0.16 | 827 | 0.01 | 57181 | 0.031311 | 0.085 | 0.21665 | 0.051 | 0.050206 | 0.056 | 198.856 | 199.12 | 203.5 | 1.5 | 203.5 | 1.5 | 0.65 | 2.34 |
| b7* | 2.42 | 68.4 | 0.17 | 393 | 0.01 | 27202 | 0.031287 | 0.28 | 0.21592 | 0.120 | 0.05007 | 0.24 | 198.70 | 198.51 | 197.4 | 5.5 | 197.4 | 5.5 | 0.53 | -0.60 |
| b9* | 4.17 | 118 | 0.14 | 842 | 0.01 | 58235 | 0.031280 | 0.10 | 0.21649 | 0.081 | 0.050219 | 0.044 | 198.66 | 198.98 | 204.1 | 1.3 | 204.1 | 1.3 | 0.86 | 2.73 |
| b11* | 0.75 | 21.3 | 0.28 | 76 | 0.01 | 5281 | 0.031268 | 0.39 | 0.21515 | 0.130 | 0.04993 | 0.33 | 198.59 | 197.87 | 190.6 | 7.7 | 190.6 | 7.7 | 0.55 | -4.15 |
| b12* | 3.97 | 113 | 0.13 | 870 | 0.01 | 60188 | 0.031393 | 0.19 | 0.21708 | 0.180 | 0.050175 | 0.044 | 199.37 | 199.48 | 202.1 | 1.3 | 202.1 | 1.3 | 0.96 | 1.39 |
| sm_b1 | 0.32 | 9.11 | 0.55 | 16 | 0.01 | 1158 | 0.031054 | 0.79 | 0.2136 | 0.110 | 0.04992 | 0.69 | 197.25 | 196.6 | 190 | 16 | 190 | 16 | 0.91 | -3.68 |
| sm_b2 | 0.30 | 8.44 | 0.38 | 22 | 0.01 | 1575 | 0.031173 | 0.50 | 0.2145 | 0.074 | 0.04994 | 0.45 | 197.99 | 197.36 | 191 | 10 | 191 | 10 | 0.75 | -3.58 |
| sm_b3 | 0.14 | 3.82 | 0.35 | 11 | 0.02 | 772 | 0.031128 | 0.86 | 0.2161 | 0.088 | 0.05036 | 0.80 | 197.71 | 198.6 | 211 | 19 | 211 | 19 | 0.69 | 6.26 |
| sm_b4 | 0.14 | 3.99 | 0.49 | 8 | 0.02 | 586 | 0.031234 | 1.1 | 0.2165 | 0.100 | 0.05030 | 1.0 | 198.37 | 199.0 | 208 | 24 | 208 | 24 | 0.69 | 4.64 |
| sm_b4-2 | 0.14 | 3.99 | 0.49 | 8 | 0.02 | 586 | 0.031336 | 1.1 | 0.2172 | 0.110 | 0.05030 | 1.0 | 199.01 | 199.6 | 208 | 24 | 208 | 24 | 0.68 | 4.34 |

data produced at the University of Geneva; * previously reported in Callegaro et al (2017)
b Measured ratio corrected for fractionation and spike contribution only.

a Corrected for initial Th/U disequilibrium using radiogenic 208Pb and a Th/U[magma] of 2.2
c Measured ratios corrected for fractionation, tracer, blank and, where applicable, initial common Pb.



**Table 4:** Compilation of all known baddeleyite (Bad)-zircon (Zrn) intergrowth types, and guideline for their interpretation.

| appearance in BSE image | Type | Common textural features | Age relationship | Rock types | References |
|---|---|---|---|---|---|
| (a) Bad / Zrn / 10 μm | igneous Bad replaced by metamorphic Zrn | frosty or raspberry-like Zrn rims or feather-like Zrn coronas; pseudomorphism; irregular crystal interfaces | Bad > Zrn | high- and low-grade meta-igneous rocks | e.g., Heaman and LeCheminant (1993); Söderlund et al. (2013); this study |
| (b) Bad / Zrn | late igneous Zrn rim on igneous Bad | euhedral Zrn rims; straight interfaces with Bad | Bad > Zrn ($\approx$) | igneous rocks which record the overstep of $SiO_2$ oversaturation | e.g., Renna et al. (2011) |
| (c) Bad / Zrn / 10 μm | igneous Bad with xenocrystic Zrn inclusions | complete Bad mantle around Zrn | Bad < Zrn | gabbros; probably other igneous rocks | this study |
| (d) Zrn / Bad | Bad with Zrn rim closed to impact melt pockets | Bad often deformed, with degraded crystallinity or disintegrated into granular droplets; Zrn rims discontinuous up to few μm | Bad > Zrn | meteorites | Moser et al. (2013); Darling et al. (2016) |
| (e) Zrn / Bad | Zrn decomposition during impact melting | droplets of Bad or other $ZrO_2$ polymorphs; Zrn often with deformation features similar to Bad in (d) | Bad < Zrn | impact glasses | e.g., El Goresy (1965); Wittmann et al. (2006) |
| (f) Bad / Zrn | desilification Bad on mantle-derived Zrn | feather-like Bad rim (often intergrown with diopside) on a Zrn megacryst | Bad < Zrn ($\approx$) | kimberlites | e.g., Kresten (1973); Heaman and LeCheminant (1993) |
| (g) Bad / Zrn | altered igneous Zrn with secondary Bad inclusions | Bad often arranged along the Zrn zonation or along cracks | Bad < Zrn | altered igneous rocks, including siliceous rocks | e.g., Lewerentz et al. (2019); this study |