# Peer review of "Multi-method U-Pb baddeleyite dating: insights from the Spread Eagle Intrusive Complex and Cape St. Mary's sills, Newfoundland, Canada."

_Geochronology, 2019_

## Referee Comment (RC1) · Ulf Söderlund (Referee) · 11 Mar 2020

This manuscript deals with an important and interesting topic, i.e. the mechanisms causing discordance of the U-Pb system in baddeleyite. I fully agree with the authors that understanding these controls are fundamental for the interpretation of discordant U-Pb data sets, and ultimately, for accurate age interpretation. This study combines age data (ID-TIMS and SIMS) of two samples with high-resolution imaging (BSE- and

[Figure]

CL-imaging) of baddeleyite in rocks that are variably affected by secondary processes (metamorphism). In addition to discussing causes of discordance, the ms includes a compilation of hitherto known baddeleyite-zircon intergrowth relationships.

I regret to say that although this manuscript embraces many complexities causing discordance, it does not provide definite answers to any of these problems. In addition to a no. of "geological causes" such as diffusional Pb loss, mixing between bd/zrn of different generations, isotopic disequilibrum, etc. – there also exist analytical complications that could trigger "apparent" discordance such as matrix effects, crystal orientation effects, mass fractionation, etc. (all of variably importance depending on what technique is applied – SIMS, TIMS, LA-ICPMS). I find this study too ambitious in its aim to deal with far too many of these complexities. The ms would benefit from instead focusing on one or two, starting with careful selection of suitable samples depending on what mechanism(s) to be study. The ambition to address all these complexities of data from complex (partly metamorphosed) samples using both SIMS and ID-TIMS makes it difficult to grasp and interpretations/conclusions not convincing. Sorry to say, but the ms leaves the reader with a bunch of questions unanswered. I agree that "microscale imaging is powerful for extracting reliable age information from complex baddeleyite grains", but that is something I would say most geochronologists already would agree on.

Indeed, the topic is important and interesting, but in order to make real progress each one for these mechanisms requires to be dealt with rigorously, and preferrably "one-by-one". With such a strategy, selection of optimal samples is crucial. For instance, if one want to investigate discordance related to matrix effects of SIMS data on baddeleyite, the samples should be top-quality, "simple" baddeleyite grains with no trace of secondary affects (i.e. 2nd zircon, alteration features, etc.). If choosing high-quality grains from rocks of different age (that could include bd reference samples), such study could also address the effect of oxygen flooding depending on crystal orientation. Perhaps the authors would then be able to identify a threshold with respect to age of sample

when 206Pb/238U dates are preferred over 207Pb/206Pb dates. On the other hand, if you want to deal with discordance related to metamorphism, then it would be advantageous to exactly know the age of metamorphism, and preferably work on samples with large time differences between protolith age and age of metamorphism.

The samples studied here have protolith ages of ca. 500 Ma and 440 Ma, and which have "experienced deformation and pervasive low-grade metamorphism lasting from ca. 420-360 Ma". This complicates the interpretation of lower intercepts and the reasoning about when and what mechanism(s) caused discordance. Figure 12 indeed shows the importance of choosing the right samples in this respect. If disregarding causes linked to "isotopic equilibrium" - which I personally believe are of less importance in general compared to other mechanisms – any analysis will plot in the triangular, grey area shown in Figure 12. I am sure the authors agree on that in order to evaluate the relative importance of controls causing discordance one should select samples that have significant age differences with respect to "protolith age", "metamorphic event" and "recent Pb loss". The samples chosen for this study do not fulfill these criteria.

I wish I could be more positive but I cannot recommend this manuscript for publication in its present shape. I still think there are some good "pieces" that can, and should, be saved/published. The obtained crystallization ages of these intrusions are overall robust and I would suggest the authors to considering publish these in a more regular "geological" journal. Since I cannot recommend publication of the submitted ms I have not made any detailed review of text and structure. Nevertheless there are a few issues I would like to comment on and that I hope could be helpful to the authors in future work:

1. For two samples, FP6D monzonite and S2E granophyre, the authors discuss the negative lower intercepts possibly reflecting remobilization of 222Rn. From my experience negative lower intercepts is something one see very rarely. The lower intercept of S2E is -229 +/- 370 Ma, thus within 0 Ma given the uncertainties so you cannot really state the l.i. is negative for that particular sample within stated uncertainties. The negative l.i. age of FP6D is largely controlled by fraction #11. I recalculated that sample, and if removing that analysis in the regression one still end up with a negative l.i. but then very close to "embrace" 0 Ma. You may be right, but more evidence is required.

2. Likewise I am not convinced about the interpretation of zn-bd intergrowth in one of the samples (FP12, SEIC?), i.e. zircon cores surrounded by baddeleyite are xenocrysts. From a textural viewpoint, it seems that many zn-bd intergrowths are sometimes very complex with "irregular" boundaries between bd and zn. Can you be sure this core-rim relationship is not apparent?, i.e. the result of a cutting affects and complex intergrowth? As argument you claim the zn cores are older than the rim, which truly would justify the zn cores to represent xenocrysts. Figure 10 shows one of these zn-bd grains. I agree the cores seem to have older 206Pb/238U dates, but here comes the difference between "age" and "date" into play. I doubt the zircon analyses have significantly older 207Pb/206Pb dates? Looking at the SIMS data on standard samples (SL18, Figure 9), the 206Pb/238U are not always reproducible, at least not from in-situ analyses. Possible biases related matrix-effects from these complex grains in the SEIC sample(s) should be even greater for composite grains, yielding 206Pb/238U that well could be biased towards older dates. Finally, I have problems imaging the process. Baddeleyite that forms in igneous systems (i.e. from Si-poor magmas) requires that magmas eventually reached Zr-saturation. This is why Bd is always (with rare exeptions) found in interstitial volumes representing the last % of liquid. Why and how would zircon xenocrysts remain in the final liquids without being trapped as inclusions in early feldspars and Fe-Mg phases?

3. About Section 6.4. "Approaches to obtain the most accurate baddeleyite crystallization ages"- I would like to make a general comment, since these type of discussions are often seen in publications. Typically, these discussions circle around the importance of high-resolutions techniques, understanding discordance, micro-scale imaging, etc. etc. However, if we really want to make improvements and find out what really matters for "obtaining accurate ages", then I infer that all these contributions fail to recognize/address the most important approach.

I have separated baddelyite from plus 500 samples over the last decades. I dear to say, that there is an almost perfect (!) correlation between quality of samples (fresh, pristine igneous mineralogy) and quality of baddeleyite grains. . . . and thus, concordant data. I am confident that at least two of the co-authors of this ms would agree on that the most important "approach to obtain accurate ages" would be if we spent only a bit more time in the field to find the best, coarsest, most pristine, and the least altered sample. Careful petrographic studies on thin sections would additionally be favorable in order to identify the most suitable samples. So, if the authors in future manuscripts want to discuss "approached to improve accurate U-Pb baddeleyite ages", then do not forget to highlight that selection of sample for processing should be given highest priority.

Despite the authors may find this review "harsh", I hope they find my comments and decision justified. They are welcome to contact me directly if something is unclear or they want further inputs.

Lund 2020-03-09 Best regards, Ulf Söderlund

---

## Referee Comment (RC2) · Anonymous Referee #2 · 13 Mar 2020

The manuscript "Baddeleyite microtextures and U-Pb discordance: insights from the Spread Eagle Intrusive Complex and Cape St. Mary's sills, Newfoundland, Canada" dealt with an very important issue about the U-Pb age discordance of Baddelyite and implications for the U-Pb age interpretation. The authors provided very detailed petrological and minerological evidences for the occconduct urence of the baddelyite and then conducted SIMS and TIMS analyses of the U-Pb ages to discuss the possible mechanisms for U-Pb discordance and to constrain the ages of the studied samples. However, there are several weaknesses about the manuscript at its present style including data precise and geological interpretationïijŃand I can't recommend it to be accepted at present version. Major comments: 1. The authors presented different sessions for the SIMS U-Pb analyses, which are the important basis for the further discussion. The data are not in enough high quality to do such things, including the discordance U-Pb ages and the inheritence of xenocrystic zircons. The precisions of some data are even lower than those reported in 1993. The selection of 206Pb/238U or 207Pb/206Pb ages to represent the studied samples are very arbitrary. The high common Pb abundances are also strange for most zircon and baddelyite grains that have high U- contents, which might be resulted from the analyses of the alteration domains. For samples FP6D and S2E, the 207Pb/206Pb ages are essential the same within the analytical errors and could be used with caution to discuss the linear correlation with the percentage of discordance. 2. The interpretation of secondary baddelyite and xenocrystic zircons are not very solided based on the presented evidences: What is unique for sample S2C to transform zirocn into secondary baddelyite under low metamorphic conditions, which should be clear addressed. In sample FP12A, the 206Pb/238U ages are not precise enough to drawimportant conclusion for such an unreported phenomenon; No resorption textures can be observed to support the authors interpretation. 3. The ages of the studied samples are not refined from the present study but mostly cited from previous result to the selection.

---

## Author Comment (AC1) · 17 Apr 2020

Our manuscript (MS) "Baddeleyite microtextures and U-Pb discordance: insights from the Spread Eagle Intrusive Complex and Cape St. Mary's sills, Newfoundland, Canada" has been reviewed by Ulf Söderlund (RC1) and an anonymous reviewer (RC2), to whom we like to express our gratitude. They perceived several weaknesses in the current form of the MS, to which we respond below (RC1 and RC2 concern largely similar aspects, therefore we address both in a single response).

[Figure]

Before addressing the major points of criticism in detail, we wish to clarify a potential misunderstanding of our intentions regarding this MS. RC1 is positive about the data and their geologic significance, but critical about the scope of the MS. The title, abstract and introduction of the original MS provocatively emphasized baddeleyite discordance as its main focus, because this is an unresolved problem for many applications of baddeleyite geochronology. We agree that this can be perceived as overreaching. Originally, the project started as a case study to constrain intrusion ages of mafic dikes and sills in Newfoundland, with the side topic of obtaining some methodical insights. However, due to the complexity of the obtained dates, addressing causes of baddeleyite discordance became an essential issue. We agree with RC1 that the selection of the samples studied here was not suited for addressing such a broad topic that challenges baddeleyite geochronologists already for a long time. We nonetheless believe that lessons learned from our case study make this a worthwhile contribution within the scope of the journal. To avoid future misunderstandings of our intentions, we propose to change the title into "Multi-method U-Pb baddeleyite dating: insights from the Spread Eagle Intrusive Complex and Cape St. Mary's sills, Newfoundland, Canada". This indicates the paper as more than simply a regional study, but removes any inference that we will resolve baddeleyite discordance completely. Our discussions about baddeleyite discordance based on our data will remain in the paper, but we will recast their presentation to clarify that they are necessary complexities, not the primary goal.

RC1, C2: "I find this study too ambitious in its aim to deal with far too many of these complexities. The ms would benefit from instead focusing on one or two, starting with careful selection of suitable samples depending on what mechanism(s) to be study."

We agree that a systematic study of baddeleyite discordance should be planned in the way proposed by RC1. Nonetheless, we think that some of the complexities we encountered in our study are of broader significance. Thereby, we propose to change the title and rewrite certain paragraphs to downplay explaining discordance of baddeleyite data in general as the primary goal.

RC1, C2: "I agree that "microscale imaging is powerful for extracting reliable age information from complex baddeleyite grains", but that is something I would say most geochronologists already would agree on."

We agree with this assessment. Nevertheless, we learned a lot from imaging crystals before and after analysis, and this is not always common practice. Some of the textures in our crystals have not been previously documented, and the microtextural information is critical for the interpretation of our complex data. We will rewrite sections to avoid sounding like the use of microtextural information is a new discovery, and instead, simply focus on its application to our samples. The value of microtextural information should be evident by this approach.

RC1, C2-3: "If choosing high-quality grains from rocks of different age (that could include bd reference samples), such study could also address the effect of oxygen flooding depending on crystal orientation. Perhaps the authors would then be able to identify a threshold with respect to age of sample when 206Pb/238U dates are preferred over 207Pb/206Pb dates. On the other hand, if you want to deal with discordance related to metamorphism, then it would be advantageous to exactly know the age of metamorphism, and preferably work on samples with large time differences between protolith age and age of metamorphism."

We fully agree with RC1. We would like to emphasize that some of these aspects (e.g., the impact of oxygen flooding) have already been addressed in previous publications using suitably homogeneous baddeleyite reference materials. We therefore refer to and have cited these publications. Our new title and decreased emphasis on discordance should minimize this concern.

RC1, C3: "The samples studied here have protolith ages of ca. 500 Ma and 440 Ma, and which have "experienced deformation and pervasive low-grade metamorphism lasting from ca. 420-360 Ma". I am sure the authors agree on that in order to evaluate the relative importance of controls causing discordance one should select samples that

have significant age differences with respect to "protolith age", "metamorphic event" and "recent Pb loss". The samples chosen for this study do not fulfill these criteria."

Again, we agree with RC1. However, our study did not intend to focus mainly on this aspect, and we will change the title and emphasis to avoid this criticism.

RC1, C3: "The obtained crystallization ages of these intrusions are overall robust and I would suggest the authors to considering publish these in a more regular "geological" journal."

We maintain that the detail of microtextural analysis and the combination of in-situ and high-precision techniques is of more interest to geochronological practitioners (and hence within the scope of GChron) than just consumers of geochronological data. We believe that many valuable methodological aspects of our study would get lost if published solely targeting an audience interested in the regional geology.

RC1, C3-4: "1. For two samples, FP6D monzonite and S2E granophyre, the authors discuss the negative lower intercepts possibly reflecting remobilization of 222Rn. From my experience negative lower intercepts is something one see very rarely. The lower intercept of S2E is -229 +/- 370 Ma, thus within 0 Ma given the uncertainties so you cannot really state the l.i. is negative for that particular sample within stated uncertainties. The negative l.i. age of FP6D is largely controlled by fraction #11. I recalculated that sample, and if removing that analysis in the regression one still end up with a negative l.i. but then very close to "embrace" 0 Ma. You may be right, but more evidence is required."

RC2, C2: "For samples FP6D and S2E, the 207Pb/206Pb ages are essential the same within the analytical errors and could be used with caution to discuss the linear correlation with the percentage of discordance."

Our interpretation of preferential 206Pb loss was not reached lightly, but involved numerous considerations, such as common Pb composition and various linear regression

strategies. Our original presentation of the ID-TIMS data attempted to be concise and may not have adequately presented these considerations, leaving our conclusion open to scepticism such as expressed above. We propose to expand the presentation of ID-TIMS data to better forestall these types of objections.

It may be that 206Pb-biased loss in baddeleyite is rare, or that it is ubiquitous, but rarely evident outside of bulk (unbiased) Pb loss and analytical error. Recent high-precision U-Pb data for baddeleyite and co-existing zircon from the Duluth gabbro (Hoaglund, 2010; Ibañez-Mejia and Tissot, 2019) require either 206Pb-biased loss in baddeleyite, or excess 207Pb due to the incorporation and decay of 231Pa, and in several early studies 206Pb loss was proposed to explain differences between dates of baddeleyite and co-existing minerals (Davis and Sutcliffe, 1985; Heaman and LeCheminant, 2000). Excess 207Pb cannot explain our data from FP6D. We contend that the extreme discordance of our data, especially from FP6D, and an apparent correlation between 206Pb loss and bulk Pb loss provide an additional example of 206Pb-biased loss in baddeleyite. This is an unsettling conclusion and will likely be met with scepticism, therefore we plan to expand the ID-TIMS data presentation and discussion.

RC1's contention that the negative lower intercept of FP6D is largely controlled by fraction #11 is misleading and not accurate. Fraction #11 is the least discordant analysis (thereby has the least amount of Pb loss), it is the most precise, has one of the highest radiogenic to common Pb ratios (Pb*/Pbc of 16.8) and is arguably the most robust data point from the whole sample. Removing it from the regression is ill-advised. A better test of whether the lower intercept is negative due to correlation between bulk Pb loss and 206Pb-biased loss, is to calculate the regression of the 4 most discordant analyses, the ones with the most total Pb loss. This leads to a lower intercept of −466 ± 390 Ma, similar to the regression using all 6 analyses and negative outside of error. We will add this test to our discussion of the ID-TIMS data, along with a much more thorough presentation of potential impacts of our choice of Pb isotopic composition of the common Pb.

RC2's comment that the 207Pb/206Pb dates are within error isn't correct for FP6D, as the individual 207Pb/206Pb dates lack consistency. Both RC1 and RC2's comments point out inadequacies in our presentation, which we will try to address with additional text.

RC1, C4: "2. Likewise I am not convinced about the interpretation of zn-bd intergrowth in one of the samples (FP12, SEIC?), i.e. zircon cores surrounded by baddeleyite are xenocrysts. From a textural viewpoint, it seems that many zn-bd intergrowths are sometimes very complex with "irregular" boundaries between bd and zn. Can you be sure this core-rim relationship is not apparent?, i.e. the result of a cutting affects and complex intergrowth? As argument you claim the zn cores are older than the rim, which truly would justify the zn cores to represent xenocrysts. Figure 10 shows one of these zn-bd grains. I agree the cores seem to have older 206Pb/238U dates, but here comes the difference between "age" and "date" into play. I doubt the zircon analyses have significantly older 207Pb/206Pb dates? Looking at the SIMS data on standard samples (SL18, Figure 9), the 206Pb/238U are not always reproducible, at least not from in-situ analyses. Possible biases related matrix-effects from these complex grains in the SEIC sample(s) should be even greater for composite grains, yielding 206Pb/238U that well could be biased towards older dates. Finally, I have problems imaging the process. Baddeleyite that forms in igneous systems (i.e. from Si-poor magmas) requires that magmas eventually reached Zr-saturation. This is why Bd is always (with rare exeptions) found in interstitial volumes representing the last % of liquid. Why and how would zircon xenocrysts remain in the final liquids without being trapped as inclusions in early feldspars and Fe-Mg phases?"

RC2, C2: "The interpretation of [. . .] xenocrystic zircons are not very solided based on the presented evidences: [. . .] In sample FP12A, the 206Pb/238U ages are not precise enough to drawimportant conclusion for such an unreported phenomenon; No resorption textures can be observed to support the authors interpretation."

Regarding sectioning effects, we have long worried about this possibility ourselves.

Unless microbaddeleyite was imaged in 3D (a methodological challenge that would be well beyond our means), we cannot for certain rule this out and in fact think that this is possibly the case for some crystals imaged (most likely Fig. 4f). However, the crystals we observed (Fig. 4e-l) sometimes even lack visible metamorphic zircon overgrowths completely, and we observed this texture repeatedly in one sample, but not in others. We are aware of the strong limitations of the SIMS data in this case, and will make sure to emphasize the textural arguments in the revised MS rather than the SIMS data.

We agree with RC1 that zircon xenocrysts would theoretically be trapped in major phases or dissolved before the last percent of melt crystallizes. Nonetheless, in the MS we propose a possible mechanism to explain this observation, where localized Zr enrichment from the dissolving zircon can be reasonably expected to trigger baddeleyite saturation in a melt that is otherwise undersaturated, thus baddeleyite may crystallize earlier as usual on such xenocrysts in rare cases. Indeed, as suggested by RC1, major phases could trap these crystals, and we have possible evidence of this. For example, baddeleyite crystals with zircon interiors in Fig. 4e/10 are not from an interstitial melt pocket, but from the margins of a partially chloritized pyroxene phenocryst, suggesting that crystallization of the pyroxene may have caught this process in the act. We will add this observation to the text.

In a revised version of the MS, we would maintain presenting textural arguments for xenocrystic zircon cores and give more emphasis that this is a rare textural observation restricted to sample FP12A. This can be done rather concisely, not requiring for a separate subchapter as in the original MS. We would keep our textural compilation (Fig. 11) and add a question mark in Fig. 11c.

RC1, C5: "I am confident that at least two of the co-authors of this ms would agree on that the most important "approach to obtain accurate ages" would be if we spent only a bit more time in the field to find the best, coarsest, most pristine, and the least altered sample. [. . .] So, if the authors in future manuscripts want to discuss "approaches to improve accurate U-Pb baddeleyite ages", then do not forget to highlight that selection

of sample for processing should be given highest priority."

We fully agree with RC1 that careful sample selection in the field can reduce complexities and we will add this specifically to our list of strategies for improving baddeleyite dates. Our field work in the investigated units followed this approach. We chose the samples for dating carefully from the large sample set we collected and we will emphasize this in the revised MS. However, orogenic metamorphism, and possibly widespread hydrothermal processes following intrusion in an active rifting environment, affected baddeleyite-bearing rocks in our study area pervasively. Such circumstances are common in the geologic record, thus the results of our case study are of broader significance.

RC2, C2: "The authors presented different sessions for the SIMS U-Pb analyses, which are the important basis for the further discussion. The data are not in enough high quality to do such things, including the discordance U-Pb ages and the inheritence of xenocrystic zircons. The precisions of some data are even lower than those reported in 1993."

For addressing U-Pb discordance, our discussion is based more on the ID-TIMS data than on the SIMS data. We want to clarify that our ID-TIMS data are less precise than those of Greenough et al. (1993) because our data are from single crystals, whereas the previous data are for aliquots containing dozens of crystals. We present a number of different data sets and will try to be clearer about which data we used for which conclusions.

RC2, C2: "The selection of 206Pb/238U or 207Pb/206Pb ages to represent the studied samples are very arbitrary."

We disagree. The MS carefully outlined alternative interpretations of the data. There is a choice of "dates" (206Pb/238U, 207Pb/206Pb, concordia upper intercept, etc.) that needs to attain an "age", yet this process is not arbitrary but rather based on justifiable hypotheses regarding the causes of baddeleyite discordance. In empirical sciences,

making "wrong" choices cannot be ruled out, but we believe that we can present a convincing rationale for our preferred "age". We will attempt to make this clearer in the revised MS.

RC2, C2: "What is unique for sample S2C to transform zirocn into secondary baddelyite under low metamorphic conditions, which should be clear addressed."

This aspect will be given more emphasis in the revised version.

RC2, C2: "The ages of the studied samples are not refined from the present study but mostly cited from previous result to the selection."

We think this criticism of RC2 misses some important points: (1) our age for S2E combines data of Greenough et al. (1993) and our new analyses, lending more confidence to earlier ages; (2) data for mafic dike FP6D is the first radiometric age ever reported for the Spread Eagle Intrusive Complex (SEIC).

In summary, we see the following important implications of our MS that we will attempt to strengthen in a revised version, showing that it transcends the scope of a regional geology study:

(1) It is a comprehensive, multi-method case study of natural baddeleyite with complex intergrowth textures and considerable degrees of discordance (in contrast to RC1, we see this non-ideal behaviour not as a drawback, but as a strength of our study, because such non-ideal samples are often the rule rather than the exception).

(2) We present a detailed microtextural documentation that, to our best knowledge, shows some textures not previously documented. We may not be able to unambiguously prove the origins of some of these textures, but we give well-founded hypotheses that can lay the groundwork for future study.

(3) We show the suitability of the SIMS in-situ approach for Mesozoic samples (using sample SL18 from the CAMP); something not previously demonstrated. This is also important to bolster confidence in the data for the Paleozoic SEIC.

(4) Using our samples, we discuss some of the conceptual limitations in reliable age determination for baddeleyite crystals that show alteration features (e.g., common Pb) and/or complex intergrowths. The complementary advantages of high precision and high spatial resolution methods provide important information in such cases, but an improved understanding of discordance is also critical for accurate age interpretation.

(5) We point to two aspects of baddeleyite discordance that are worth considering by future geochronologic studies: first, that the rims of crystals appear to be more affected by Pb loss than the cores (demonstrated by comparison of SIMS spot analyses to bulk dissolution ID-TIMS data and the correlation of discordance with smaller grain size in SL18), suggesting that Pb loss is diffusional, and second, that there may be a component of preferential 206Pb loss in discordant baddeleyite data, at least in specific cases. These implications have consequences for baddeleyite dating strategies. A comprehensive treatise of all possible causes of discordance in baddeleyite is beyond the scope of this MS, but our discussion may prove helpful to direct future attempts in understanding aspects of baddeleyite discordance.

Specific changes will be made in the Introduction to better define the scope of the paper according to the points made above. The chapters "Regional geology", "U-Pb geochronology methods", and "Petrography" will only require minor modifications. In "U-Pb results", we will expand the presentation of ID-TIMS data to address some of the scepticisms of the reviewers. We will re-organize the Discussion into two main subchapters: "Occurrence, textures and interrelations of accessory minerals" and "Interpreting intrusion ages from non-ideal baddeleyite". The present subchapter "Zircon inclusions in baddeleyite" will be shortened and integrated into the first subchapter.

—

With our most sincere regards

Johannes E. Pohlner, Axel K. Schmitt, Kevin R. Chamberlain, Joshua H. F. L. Davies, Anne Hildenbrand, and Gregor Austermann

References (only those not mentioned in the MS):

Hoaglund, S. A.: U-Pb geochronology of the Duluth Complex and related hypabyssal intrusions: investigating the emplacement history of a large multiphase intrusive complex related to the 1.1 Ga Midcontinent Rift. Unpublished PhD thesis, University of Minnesota, 103 pp., 2010. Ibañez-Mejia, M. and Tissot, F. L.: Extreme Zr stable isotope fractionation during magmatic fractional crystallization. Science Advances, 5(12), eaax8648, 2019.

---

## Author Response (AR1)

**Multi-method U-Pb baddeleyite dating: insights from the Spread Eagle Intrusive Complex and Cape St. Mary's sills, Newfoundland, Canada.**

Johannes E. Pohlner1,2, Axel K. Schmitt1, Kevin R. Chamberlain3, Joshua H. F. L. Davies4,5, Anne 5 Hildenbrand1, and Gregor Austermann1

[revised manuscript text omitted]
 (206Pb/238U) and  $497.8 \pm 73.2$  Ma  $(^{207}\text{Pb}/^{206}\text{Pb})$ . For FP12A baddelevite, the weighted mean dates are 508.2 ± 11.2 Ma  $(^{206}\text{Pb}/^{238}\text{U})$  and 546.6 ± 83.6 Ma (207Pb/206Pb). Many baddeleyite analyses show surprisingly high contents of common Pb, those with <90% radiogenic 206Pb were generally excluded from weighted mean calculations. During grain mount sessions, FC-4b baddeleyite was analyzed as a secondary reference in addition to Phalaborwa baddelevite. Weighted mean 206Pb/238U dates of FC-4b calculated with 250 Phalaborwa reference are  $1118 \pm 39$  Ma (MSWD = 0.63; n = 28),  $1101 \pm 44$  Ma (MSWD = 0.41; n = 29),  $1124 \pm 56$  Ma (MSWD = 0.91; n = 9) and  $1117 \pm 23$  Ma (MSWD = 2.42; n = 18; session with sample S2E). Therefore, in all grain mount sessions, the 206Pb/238U ID-TIMS reference age of FC-4b (1096.84 ± 0.45 Ma, Schmitt et al. 2010) was reproduced within error limits. Likewise, the  ${}^{207}Pb/{}^{206}Pb$  dates we obtained for Phalaborwa (2058.8 ± 0.7 Ma, MSWD = 6.3, n = 254) and FC-4b baddelevite (1096.0  $\pm$  2.9 Ma, MSWD = 1.1, n = 84) are in good agreement with the ID-TIMS 207Pb/206Pb data (2059.6  $\pm$  0.35 255 Ma, Heaman, 2009, and  $1099.6 \pm 1.5$  Ma, Schmitt et al., 2010). Because of the consistency of Phalaborwa and FC-4b results, analyses from both reference materials were combined for obtaining the U/Pb relative sensitivity factor. Despite the good agreement of 206Pb/238U dates of the reference baddeleyite, 206Pb/238U dates of baddeleyite from sample FP6D obtained during the same sessions were less reproducible ( $516.2 \pm 21.2$  Ma,  $531.9 \pm 14.1$  Ma and  $563.4 \pm 15.2$  Ma), with reverse discordance
- 260 in sessions two and three (Figure 6b, c). However,  $^{207}$ Pb/ $^{206}$ Pb dates of these sessions are consistent (500.8 ± 18.0 Ma, 502.5 ± 18.0 Ma).

8.6 Ma and 484.1  $\pm$  13.5 Ma), yielding a total weighted mean 207Pb/206Pb date of 497.0  $\pm$  6.8 Ma (MSWD = 0.75; n = 77). Common Pb contents tend to be lower than in FP7G and FP12A, but are often still significant. Zircon rims on baddeleyite and baddeleyite-free zircon from all SEIC samples yielded a wide range of 206Pb/238U dates from 142–517 Ma (Figure 6d, f). At least for sample FP6D, 206Pb/238U dates become younger with increasing U contents. Zircon analyses from SEIC samples have

- mostly high common Pb contents (<90% radiogenic 206Pb). Zircon inclusions in baddeleyite (FP12A) yielded 206Pb/238U date ranges of 470–733 Ma and 297–607 Ma for baddeleyite- and zircon-based RSC, respectively.
  For CSMS, baddeleyite of sample S2E (Figure 7; Table 2; Table S5) yielded weighted mean dates of 446.6 ± 15.4 Ma (206Pb/238U; MSWD = 1.44; n = 21) and 436.5 ± 21.2 Ma (207Pb/206Pb; MSWD = 0.82) from the in situ session. In contrast, the grain mount session of the same sample yielded 491.0 ± 19.8 Ma (206Pb/238U; MSWD = 0.45; n = 39) and 425.5 ± 8.7 Ma
- 270 (207Pb/206Pb; MSWD = 1.00), showing reverse discordance (Figure 7b). 206Pb/238U zircon dates from grain mounts are in the range of 411–443 Ma with moderate or low common Pb contents, but in situ dates of anhedral zircon in chlorite pseudomorphs are much younger, combined with high U and common Pb contents. Zircon dates from S2C (Figure S9; Table S4) are often younger than S2E baddeleyite, but most analyses show high common Pb.

For SL18, weighted mean dates are 202.5 ± 2.2 Ma (206Pb/238U) and 182.7 ± 12.5 Ma (207Pb/206Pb) for the grain mount session

[revised manuscript text omitted]